# Revenge of the Tick: Tick-Borne Diseases and the Eye in the Age of Climate Change and Globalisation

Xin Le Ng [1,2], Berdjette Y. Y. Lau [2], Cassandra X. C. Chan [2], Dawn K. A. Lim [1,2], Blanche X. H. Lim [1,2] and Chris H. L. Lim [1,2,3,4,*]

1 Department of Ophthalmology, National University Health System, Singapore 119228, Singapore
2 Yong Loo Lin School of Medicine, National University of Singapore, Singapore 119228, Singapore
3 Singapore Eye Research Institute, Singapore 169856, Singapore
4 School of Optometry and Vision Science, UNSW Sydney, Kensington, NSW 2052, Australia
* Correspondence: chris_hl_lim@nuhs.edu.sg; Tel.: +65-6908-2222

**Simple Summary:** Zoonotic diseases, including tick-borne diseases, are posing an increasing health burden on society, even more so with climate change and increased geographical interconnectivity in the 21st century. Ocular manifestations may be observed in tick-borne diseases. Our article provides a comprehensive overview of ocular disease associated with ticks. We also aim to fill in the gaps in the existing literature by emphasising the increasing importance and prevalence of tick-borne diseases in non-endemic areas due to climate change. The findings of this review may serve to benefit medical practitioners, as we propose management strategies for patients and increase the awareness of tick-borne eye diseases in this current climate.

**Abstract:** Climate change has contributed to changes in disease transmission. In particular, zoonoses such as tick-borne diseases are occurring in areas previously unsuitable for tick survival, with spread to non-endemic areas rising. Ophthalmic manifestations of tick-borne diseases are rare. Often overlooked, diagnosis requires awareness and a high level of suspicion, which may delay treatment. This review provides a comprehensive overview of ocular disease associated with ticks so that management protocols for patients can be designed and implemented. A narrative literature review was conducted. The current literature includes case series, case reports, and literature reviews. Ocular manifestations of tick-borne diseases include adnexal manifestations, conjunctivitis, keratitis, cranial nerve palsies, optic nerve disease, uveitis, exudative retinal detachment, and panophthalmitis, which may occur in isolation or as part of a systemic process. As there is no one constellation of ocular symptoms and signs diagnostic of tick-borne eye diseases, a systematic approach is recommended with particular attention to significant travel and exposure history. In this review, we identify significant risk factors and propose management strategies for afflicted patients to improve treatment outcomes while maintaining cost-effectiveness. Ophthalmologists and generalists will benefit from increased awareness of ocular manifestations of tick-borne diseases in the age of modern travel and climate change.

**Keywords:** zoonoses; ticks; tick-borne eye disease; ocular manifestations; climate change

## 1. Introduction

Tick-borne diseases are known to manifest in the eye either in isolation or as part of a systemic disease process. This is true of most infectious diseases. The saliency of this paper lies in the lack of awareness of ticks as a potential cause of ocular disease, especially in non-endemic areas. This may contribute to late diagnosis and treatment delays.

Ticks are arachnid arthropods that can be found worldwide. They serve as vectors for a variety of diseases. The infected tick transmits bacterial pathogens via tick bites, which then enter the bloodstream of hosts. Some species of ticks thrive in grassy wooded areas

accounting for a higher incidence of tick-borne diseases in rural environments. Urbanisation and increased global interconnectivity, however, has increased the prevalence of tick-borne diseases [1]. Furthermore, with climate change, there is evidence for an expanded range in which some tick species can be found that were not previously known to be endemic for them [2,3].

In this article, we look at ocular manifestations of the following as listed in Table 1 [4–7].

**Table 1.** Table relating the tick-borne disease with its respective aetiological and transmission agents.

| Disease | Aetiological Organism | Transmission |
|---|---|---|
| Borreliosis | *Borrelia burgdorferi sensu lato* | Several tick species of the genus *Ixodes*. |
| Tularaemia | *Francisella tularensis* | Ticks of the species *Dermacentor variabilis* (Say), *Dermacentor andersoni* (Stiles), *Amblyomma americanum* (Linnaeus) |
| Babesiosis | Intraerythrocytic protozoa of the order Piroplasmida, family Babesiidae and genus *Babesia* | *Ixodes scapularis* |
| Ehrlichiosis | Intracellular Gram-negative bacteria of genus *Ehrlichia* | *Amblyomma americanum* and *Dermacentor variabilis* |
| Rickettsiosis | Obligate intracellular bacteria groups of the genus *Rickettsia Orientia tsutsugamushi* | Various tick species |
| Toxoplasmosis | *Toxoplasma gondii* | Tick species of the genera *Ixodes*, *Dermacentor*, *Amblyomma* |
| Tick-borne Encephalitis | Tick-borne encephalitis virus, a member of the Flaviviridae family | Tick species of the genus *Ixodes* |
| Powassan Encephalitis | Powassan virus, a member of the Flaviviridae family | Tick species of the genus *Ixodes* |
| Colorado Tick Fever | Coltivirus, a member of the Reoviridae family | Tick species of the genus *Dermacentor* |
| Tick-borne Relapsing Fever | Bacteria species of the genus *Borrelia* | Tick species of the genus *Ornithodoros* |

Bacterial tick-borne illnesses often respond to antibiotics if recognised promptly. In this article, we focus on ocular manifestations of tick-borne diseases. With the increased prevalence of ticks, our review seeks to increase awareness of the diagnosis as a differential in patients with certain risk factors to facilitate prompt recognition and treatment.

## 2. Methodology

For this narrative literature review, a computerised database search was performed for relevant articles encompassing available literature on tick-borne diseases with ophthalmic manifestations on PubMed, Google Scholar, MEDLINE, and EMBASE. The search comprised the following keywords: Lyme, Borreliosis, Babesiosis, Ehrlichiosis, Tularaemia, Tick-borne Relapsing fever, Toxoplasmosis, Rocky Mountain Spotted Fever, Mediterranean Spotted Fever, Colorado Tick Fever, tick-borne diseases, uveitis, ocular complications, and ocular manifestations. Search results were screened for relevance and references cited within the articles were used to further augment the search. This review encompassed an international search, but only articles published in English were used. We restricted our search to articles published up to 31 August 2022.

### 3. Results

A total of 173 articles reported ocular manifestations of tick-borne diseases. The results are presented below and categorised into generalised ocular manifestations from various tick species and individual tick-borne diseases.

Table 2 relates the ocular manifestations and site of ocular involvement to the various tick species involved in the transmission of the tick-borne eye diseases discussed.

**Table 2.** Summary table relating the ocular manifestations and site of ocular involvement to the various tick species.

| Anatomical Involvement | Tick Species | Ocular Manifestation |
|---|---|---|
| External ocular and adnexal involvement | *I. nipponensis* *I. scapularis* *D. variabilis* *D. andersoni* *R. sanguineus* *A. americanum* | Eyelid ulceration, eyelid inflammation (acute or chronic), eyelid oedema, painful eyelid nodule ± mucopurulent discharge, palpebral ptosis, erythema migrans on lids and periocular adnexae, palpebral oedema, blepharospasm, orbit periostitis, periorbital oedema, orbital myositis (medial rectus, inferior rectus, lateral rectus) associated with lacrimal gland enlargement, dacryoadenitis, dacryocystitis, periorbital ecchymosis, conjunctival follicles with mucous discharge |
| Anterior segment | *I. ricinus* *I. persulcatus* *I. scapularis* *I. pacificus* *D. variabilis* *D. andersoni* *D. reticulatus*, and *H. longicornis* *R. sanguineus* *A. americanum* *A. cajennense* complex, mainly *A. sculptum* *O. hermsi* *O. parkeri* *O. turicata* *O. moubata* | Non-granulomatous anterior uveitis, conjunctival injection, conjunctival papule, conjunctival ulcer, conjunctival nodule, conjunctival hyperaemia, corneal precipitates, corneal thinning and vascularisation, cells and flare of anterior chamber, follicular conjunctivitis, symblepharon, subconjunctival haemorrhages, episcleritis, scleritis, keratitis (exposure, interstitial, peripheral ulcerative, stromal), Cogan's syndrome, Horner's syndrome, Argyll-Robertson pupil, afferent pupillary defect, tonic pupils, iritis, cyclitis (acute and chronic), iridocyclitis, uveitis (anterior, intermediate, posterior, panuveitis—granulomatous, associated with anterior synechiae), posterior synechiae, pars planitis, cataract formation, secondary glaucoma, band keratopathy, corneal ulcers (ameboid-type), ciliary injection, corneal oedema, corneal inflammatory infiltrates, mild infiltration of the anterior stroma, endophthalmitis, conjunctival chemosis, hypopyon, conjunctival vasculitis |

**Table 2.** *Cont.*

| Anatomical Involvement | Tick Species | Ocular Manifestation |
|---|---|---|
| Posterior segment | *I. ricinus*<br>*I. persulcatus*<br>*I. scapularis*<br>*I. cookei*<br>*I. marxi*<br>*I. spinipalpis*<br>*D. variabilis*<br>*D. andersoni*<br>*D. reticulatus*<br>*R. sanguineus*<br>*A. americanum*<br>*A. cajennense* complex, mainly<br>*A. sculptum*<br>*O. moubata*<br>*H. longicornis* | Haemorrhages (intraretinal, disc with venous engorgement, white-centred retinal, subretinal, flame-shaped), vitritis, vitreous haze and cells, focal retinitis with surrounding retinal and macular pigmentary changes, multifocal retinitis, cells and flare of vitreous cells, retinal vasculitis, posterior uveitis (focal retinochoroiditis), serous macular detachment, retinal detachment, choroiditis (chorioretinitis, multifocal choroiditis, Birdshot chorioretinopathy, acute posterior multifocal placoid pigment epitheliopathy), inflammatory choroidal neovascular membrane, vitritis (anterior 'spiderweb' vitritis without retinal involvement), vitreous clouding, atypical Eales disease syndromes, exudative retinal detachments, branch retinal artery occlusion with cotton wool spots, branch retinal vein occlusion, chorioretinal horseshoe-shaped retinal tear with inflammatory nodules on the flap, chorioretinal inflammatory foci, secondary retinitis pigmentosa, pigment epitheliitis, macular star, macular oedema, secondary glaucoma, full thickness macular hole, peripapillary retinal pigment epithelium detachments, big blind spot syndrome, optic disc staining, juxtavascular white retinal lesions, focal vascular sheathing, multiple arterial plaques, retinal vascular leakage, delayed filling in a branch retinal vein, multiple hypofluorescent choroidal dots, endophthalmitis, retinal nerve fibre layer infarct, retinal oedema, venous tortuosity, choroidal vasculitis, epiretinal membrane |
| Optic neuropathy | *I. cookei*<br>*I. marxi*<br>*I. spinipalpis*<br>*I. scapularis*<br>*I. ricinus*<br>*D. variabilis*<br>*D. andersoni*<br>*D. reticulatus*<br>*R. sanguineus*<br>*A. americanum*<br>*A. cajennense* complex, mainly<br>*A. sculptum*<br>*O. hermsi*<br>*O. parkeri*<br>*O. turicata*<br>*O. moubata*<br>*H. longicornis* | Neuroretinitis, papillitis (optic neuritis), optic atrophy secondary to optic nerve involvement, cranial nerve palsies (CN 3, 4, 6, 7), paralytic strabismus, acute anterior ischaemic optic neuropathy, optic perineuritis, papilloedema, optic nerve sheath contrast enhancement, disc oedema, optic nerve oedema |

*3.1. Adnexal Lesions*

3.1.1. Eyelid Nodules (Present in the Same Location after the Tick Bites)

Lai et al. reported a case of a three-year-old girl presenting with a lower lid ulcer involving an *Ixodes* tick after outdoor activities [8]. The tick body was grasped with toothed forceps and its hypostome disengaged from the skin surface. The patient was treated with 0.3% tobramycin ointment and subsequent discharged. Lee et al. described another case with an *Ixodes* tick in a 79-year-old female involving the upper lid [9]. The patient was prescribed 100 mg of doxycycline for a week as prophylaxis after removal of the tick and recovered without complications. Uzair et al. described the removal of a *Dermacentor* tick attached to the upper lid margin of a 46-year-old female using toothless forceps [10]. No details regarding pharmacological therapy were included in the case report. Shrestha et al. opted to treat their patient with topical chloramphenicol ointment and dexamethasone sodium phosphate twice daily for two weeks post-removal with the patient making a full recovery [11].

Some tick bites can go undetected for months. Rai et al. describes a case of a 75-year-old male presenting with persistent upper lid growth without inflammation that was excised under local anaesthesia [12]. Laboratory examination revealed remnants of a tick of unidentified species from the incomplete removal of a tick by the patient six months prior after a walk in a wooded area (Cape Cod, Massachusetts, United States of America) with his fingers. Histopathological analysis revealed surrounding areas of acute and chronic granulomatous inflammation in the surrounding dermis. Prior to onset of the growth, the patient had been prescribed a single dose of oral doxycycline 200 mg as Lyme prophylaxis. He made an uneventful recovery post-excision.

A large number of case reports regarding tick bites of ocular adnexa exists [11,13–17]. It is evident that a variety of tick species in various geographical locations, including urban cities such as Singapore [8], are involved in this presentation [11,13,18–20]. There have been marked variations in removal technique and subsequent treatment and follow-up.

3.1.2. Surface/Conjunctival Lesions

Kuriakose et al. describes an *Ixodes* tick masquerading as a conjunctival lesion [21,22]. A Caucasian female patient in her late 60's presented with a three-week history of a right pigmented conjunctival lesion associated with foreign body sensation that began acutely while camping in the country (Adirondacks, Upstate New York, United States of America). A tick was found on the nasal conjunctiva with conjunctival injection and prominent episcleral vessels. The tick was removed and identified as an *Ixodes* species and the patient was treated with topical polymyxin-trimethoprim three times daily and topical loteprednol twice daily for three days each. On identification of the tick, 100 mg oral doxycycline was commenced for 14 days for Lyme prophylaxis. The patient made an uneventful recovery. Kanar et al. have demonstrated the utility of spectral domain optical coherence tomography (SD-OCT) in determining the extent of the tick bite on the ocular surface as well [23], although this investigation is not routinely used in other case reports reviewed.

*3.2. Ophthalmic Lyme Disease*

Lyme disease is the most common vector-borne disease in the United States [24]. It is transmitted by the *Ixodes* tick and involves infection by the *Borrelia* bacterium, of which the most common is *Borrelia burgdorferi*, along with *Borrelia garinii* and *Borrelia afzelii*. Ocular manifestations of borreliosis are varied and extensively reviewed [25–35]. They are generally classified into three stages with no clear time delineation between stages (Figure 1) [36–38].

**Stage I: Early localised disease**
- Conjunctivitis
  - Follicular

**Stage II: Early disseminated disease**
- Uveitis
  - Anterior
  - Intermediate
  - Posterior
  - Endophthalmitis
- Optic neuropathy
  - Optic neuritis
  - Anterior Ischaemic Optic Neuropathy
  - Optic atrophy

- Papilloedema
- Optic perineuritis
- Pseudotumour cerebri
- Leber's stellate neuroretinitis
- Cranial nerve palsy
  - CN VII
  - CN III
- Pupillary disorder—however anecdotal—Horner's, Argyll Robertson

**Stage III: Late disease**
- Keratitis—interstitial/stromal
- Conjunctivitis, Episcleritis
- Orbital myositis
- Multiple cranial neuropathy

**Figure 1.** Ocular manifestations of Lyme disease according to stage [36,39,40].

Ocular manifestations may be self-limited and resolve without residual sequelae, provided the recommended antimicrobial regimen has been instituted [41]. However, ocular manifestations despite treatment are not unheard of.

3.2.1. Ocular Surface: Follicular Conjunctivitis, Episcleritis, Scleritis

Follicular conjunctivitis that is nonspecific may occur in as many as 10% of patients during the early stages of Lyme disease [42,43]. This may be associated with episcleritis, periorbital oedema, photophobia, and subconjunctival haemorrhage. Episcleritis and symblepharon have also been described [44]. Conjunctivitis and episcleritis in early stage Lyme disease are often mild and self-limited, not requiring topical antibiotics. Scleritis has also been described in conjunction with intraocular inflammation [45].

### 3.2.2. Keratitis

Keratitis is a well-documented ocular manifestation of Lyme disease [39,40,46–48]. Stromal keratitis [25,31,33,44,46,49–51] occurs late in Lyme disease and has been characterised by the appearance of superficial and deep, hazy corneal infiltrates without Descemet membrane involvement [52]. The entire stroma may be involved with scattered focal nummular opacities. Patients tend to present with mild blurring of vision and occasional photophobia [33,39,46]. Lyme keratitis is often a bilateral presentation; however, unilateral keratitis has been reported [40]. The pathophysiology is attributed to an immune response by the host during later stages of the disease. Additionally, in support of an immune mechanism, peripheral ulcerative keratitis with peripheral stromal oedema and mild corneal neovascularization has been described as related to Lyme disease [53]. Topical steroids have resulted in successful treatment of keratitis [46,48,54] further supporting an immune response as the mechanism for the keratitis. Systemic antibiotics for the underlying disease are of synergistic benefit [47]. Compared to syphilitic keratitis, Lyme keratitis is often milder with infrequent corneal neovascularization [52].

### 3.2.3. Intraocular Inflammation

Ocular inflammation [55–65] involving almost every part of the eye has been associated with Lyme disease: Iritis [29], pars planitis, vitritis, choroiditis [66,67], chorioretinitis [68], and panophthalmitis [52,54]. Johnson et al. demonstrated that early in the disease, spirochaetes disseminate throughout the body, including eyes of murine models [69,70]. It is possible that these spirochaetes are then sequestered immunologically and remain dormant. Whether reactivation and later manifestations of ophthalmic Lyme disease represents an immune privilege of *Borrelia* or a maladaptive host response remains to be seen. Preac-Mursic et al.'s demonstration of *Borrelia burgdorferi* from an iridectomy specimen in a patient presenting with anterior uveitis lends credence to *Borrelia*'s direct involvement in uveal inflammation [71].

Neuroretinitis with Lyme disease has been reported in patients though a causal relationship has not been established [30,37,72,73]. Full recovery has been described after treatment with systemic intravenous ceftriaxone [73].

Choroiditis [67] with exudative retinal detachment is associated with Lyme meningitis as well, with rapid response to treatment with doxycycline. However, as with neuroretinitis, a causal relationship is hard to establish given choroiditis is a nonspecific immunologic reaction to various infectious agents. Variations of choroiditis including chorioretinitis [74,75], multifocal choroiditis [66], multifocal placoid pigment epitheliopathy [76,77], and birdshot chorioretinopathy [78] have been described in patients with positive Lyme serologies [52].

Retinal vasculitis is associated with Lyme [79]. It was described in a case series of seven patients by Leys et al. with a range of presentations from acute visual loss to chronic uveitis [80]. Similarly, Smith et al. described three patients seroreactive for Lyme borreliosis presenting with vasculitis that could not be explained by other causes [81]. With tetracycline therapy, improvement of the vasculitis was demonstrated.

Zierhut et al. described a case of panuveitis as well; however, it was in a patient whose laboratory tests were positive for syphilis and tentative for Lyme disease [82].

Despite treatment, ocular complications may sometimes progress. In 1985, Steere et al. reported the case of a 45-year-old female from Westchester County, New York who presented with acute iritis despite treatment for systemic Lyme borreliosis with intravenous cefazolin 6 g/day for seven days. She had persistent fever despite antibiotics which was oralised to 500 mg tetracycline four times a day for seven more days after which the fever

resolved. Two weeks later, she developed acute iritis with posterior synechiae. Despite topical, subconjunctival and systemic corticosteroids (oral prednisone, 60 to 100 mg/day), her visual acuity worsened (no data on VA) and she developed panophthalmitis despite treatment with intravenous methicillin 12 g/day and intravenous gentamicin 80 mg Q8H. She eventually underwent lensectomy and vitrectomy with the administration of intravitreal gentamicin 0.2 mg and chloramphenicol 0.2 mg. However, even after a repeat vitrectomy a week later, the eye became phthisical. Pathologic examination of the enucleated eye revealed Lyme disease spirochaetes in the vitreous debris [83]. With advancements in antimicrobials since then, the new standard of treatment for Lyme disease is included in the discussion.

### 3.2.4. Chronic Lid Inflammation

Murillo et al. describes a case of a 16-year-old girl with a four-year history of recurrent eye inflammation with lid oedema, lid erythema, ptosis, and superficial venous tortuosity, associated with conjunctival hyperaemia, vascularization and thinning of the superior cornea and corneal precipitates. She was diagnosed with peripheral ulcerative keratitis [84]. Eyelid biopsy demonstrated spirochaetes and vasculitis, which was linked to Borreliosis (diagnosed by immunoblot), by which dermatoborrelioses are known.

### 3.2.5. Orbital Inflammation

Orbital myositis has been described in a few case reports [85–87]. Sauer et al. reported two female patients aged 13 and 68 years old who both lived in an area endemic for Lyme disease. The 68-year-old presented with recurrent acute right orbital swelling for three years, each episode lasting two to four weeks, two to four times a year, and each episode treated with steroids, non-steroidal anti-inflammatory drugs, or resolved spontaneously. Only when she presented with the complication of horizontal diplopia was further workup pursued and Lyme disease confirmed based on history of erythema migrans and positive *Borrelia* antibodies. Myositis was confirmed by magnetic resonance imaging (MRI). She was then treated with 200 mg oral doxycycline a day with resolution of symptoms in three weeks and no further relapses. Similarly, the 13-year-old patient was worked up for Lyme after presenting with unilateral orbital swelling and diplopia and successfully treated with four weeks of 200 mg/day oral doxycycline [88].

A case of subacute painless ptosis was reported by Xu et al. in a 90-year-old with history positive for tick bite 6 months prior to presentation and with positive serologies. Further workup noted orbital inflammation involving extra ocular muscles with lacrimal gland enlargement and contrast enhancement of optic nerve sheath on MRI [89].

### 3.2.6. Neuro-Ophthalmic Manifestations and Neuroretinitis

Cranial nerve palsies of III, IV [90,91], VI [73,92], and VII [93,94] have been reported in Lyme disease [95–99], the most common of which is cranial nerve VII palsy [52,73,100,101], which may accompany meningitis. Rarer manifestations include paralytic strabismus [102]. Often associated with other neurologic abnormalities, these palsies can occur together, with most resolving within two weeks to five months from onset.

Reversible Horner's syndrome [36,73,103,104] has also been described in Lyme disease by Glauser et al. Although initial Lyme serologies for the case were negative, they were positive two weeks post-presentation. The anisocoria resolved after a 10-day course of systemic ceftriaxone. Neuroimaging of the brain and spine with magnetic resonance imaging and chest radiographs ruled out other probable causes [103].

Lyme optic neuropathy is rare and may occur in the absence of other central nervous system signs [59,64,105–111] and may sometimes be the first presentation [112]. One case

report of retrobulbar optic neuropathy meets the criteria proposed by Sibony et al. for optic neuritis attributable to Lyme disease [59]. Optic nerve head oedema almost always is a feature. More often, the optic neuropathy develops secondary to raised intracranial pressure that may result from meningitis [52], although papilloedema in the absence of raised intracranial pressure is possible as well [113]. The resulting optic atrophy is likely a result of meningitis or papilloedema rather than a primary optic neuropathy.

Retinal vascular occlusion in Lyme disease has been described in case reports, usually of retinal vein occlusion presenting with an acute drop in visual acuity. A case report of retinal arterial occlusion was reported by Lightman et al. [112], where a 37-year-old female presented with an acute left scotoma with no known history of tick bite or skin rash. She was found to have a left inferotemporal branch retinal artery occlusion with multiple cotton wool spots and disc oedema with focal arteriolar irregularity. In addition, she had bilateral mild vitritis, and examination of the right eye revealed right optic nerve oedema with cotton wool spots and irregular retinal arterioles. A comprehensive workup was undertaken with positive Lyme serologies and negative VDRL and TPA-ABS. After treatment with intravenous penicillin G (12 million units daily) for 10 days, her systemic complaints and both vitritis and fundus findings had resolved, though the scotoma persisted. In view of the response to antibiotics, inflammatory ocular findings and positive serologies, the authors suggested the presentation to have resulted as a complication of Lyme disease. Patients generally responded well to systemic antibiotics and tapering corticosteroids (topical and systemic).

Acute ocular motor disturbances have been described in association with Lyme disease including opsoclonus and nystagmus [114,115]. Gibaud et al. reported a case of opsoclonus in a child with neuroborreliosis. The nine-year-old girl presented with left incomplete facial palsy and aseptic meningitis. There was no reported history of tick bites, travel, or erythema migrans. The patient was found to have positive serology for *Borrelia burgdorferi* with a Western blot confirmation and intrathecal synthesis of anti-Lyme antibodies was demonstrated. Other infectious causes were excluded (including syphilis). The patient showed rapid response to treatment with 2 g/day ceftriaxone with complete resolution after three weeks of treatment [116].

Neuroretinitis [73,117–120] including interdigitation zone loss [121], has been described in patients with positive Lyme serologies. Complications of neuroretinitis include development of full thickness macular holes and peripapillary retinal pigment epithelium detachments precipitated by optic nerve inflammation and secondary macular oedema [122,123].

### 3.2.7. Temporal Arteritis

Pizzarello et al. [124] reported a case of a 71-year-old male patient who presented with acute left inferior visual field loss (visual acuity 20/50 or 6/15) with temporal artery biopsy-proven giant cell arteritis. Initial treatment with oral prednisone failed to stabilise his vision, which deteriorated further (visual acuity hand movements). Some measure of vision was restored after treatment with intravenous Decadron 8 mg Q6H for 24 h and a tapering steroid regime starting with prednisone 80 mg/day. The patient lived in an area with endemic Lyme disease. Two months after his presentation, a Lyme disease titre was requested and blood cultures showed *Borrelia* species that was not *Borrelia burgdorferi*. Serum FTA-ABS and VDRL tests were negative. A course of 2 g/day intravenous ceftriaxone was administered for seven days with repeat blood cultures showing no growth and a second course of prednisone did not improve vision (visual acuity counting fingers at five feet). Silver stain of the temporal artery biopsy demonstrated spirochaetes consistent with *Borrelia* species [124].

### 3.3. Tularaemia and Oculoglandular Syndrome

Tularaemia is caused by the Gram-negative coccobacillus *Francisella tularensis* [5] and is transmitted to humans via infected animal blood, or insect bites from ticks, mosquitoes, and deer flies [125]. Common tick vectors are of the *Dermacentor* species and *A. americanum* [126]. Tick-borne tularaemia infections mostly occur in the summer, when tick activity is high [127]. The six subtypes of tularaemia are categorised by their route of transmission, and include ulceroglandular, glandular, oropharyngeal, pneumonia, typhoidal, and oculoglandular, the last form being the rarest (<5% of all cases). Oculoglandular tularaemia is transmitted via direct inoculation by the rubbing of eyes or ocular exposure to contaminated fluids, droplets, or aerosols [5]. Upon inoculation of the conjunctiva, a reddish-yellow papule may first form and develop into a granulomatous ulcer. Necrotic tissue filling the ulcer eventually sloughs off, marking the site of entry of *F. tularensis* [128].

Ophthalmic complaints previously reported in oculoglandular tularaemia include photophobia, excessive lacrimation or epiphora, foreign body sensation, sensation of soreness, lid oedema, floaters, crusting on eyelashes, and hyperaemia [128–132]. Ophthalmic manifestations include conjunctival chemosis, episcleritis, ptosis, purulent secretions, and periorbital oedema [129].

Parinaud's oculoglandular syndrome (POGS) refers to a unilateral granulomatous follicular conjunctivitis associated with mucopurulent discharge and painful preauricular and submandibular lymphadenopathy [125]. Systemic clinical features such as malaise and fever may be present [133]. POGS may be seen in cat scratch disease, tularaemia, sporotrichosis, and, less commonly, in syphilis, tuberculosis, or coccidioidomycosis [134]. Clinical differentiation among these aetiologies may be achieved by a careful history; a history of contact with cats is likely for POGS secondary to cat scratch disease, while a history of contact with infected animals such as rabbits, squirrels, and ticks, with necrotising conjunctival inflammation and ulceration suggest POGS secondary to tularaemia [133]. However, this is not absolute as Siniscal describes oculoglandular tularaemia in three patients who had close contact with a cat [128].

In Sivas, Turkey, where tularaemia is endemic, an 18-year-old female presenting with left conjunctivitis and painful preauricular and submandibular lymphadenopathies was worked up for several aetiological agents of POGS; her micro-agglutination titre was positive for tularaemia. She was subsequently treated with 12-h 100 mg oral doxycycline for a 10-day course and experienced a full recovery [135]. A nine-and-one-half-year-old male presented with right periorbital swelling, right conjunctivitis and exquisitely tender preauricular and submandibular lymph nodes, associated with a low-grade fever. A slit lamp examination revealed multiple yellow conjunctival nodules with necrotic ulcers on both upper and lower tarsal conjunctivae. Cultures taken from the blood, conjunctiva, and preauricular lymph node aspirate revealed the growth of *F. tularensis*, confirming oculoglandular syndrome secondary to tularaemia. The patient was initially treated with IV chloramphenicol and gentamicin, but confirmation of *F. tularensis* prompted gentamicin-only treatment for 10 days because he was already experiencing rapid clinical improvement [134]. Similarly, an 18-year-old male presented with left eyelid oedema with conjunctival injection associated with left-sided tender preauricular, postauricular, and submandibular lymph nodes. A slit lamp examination showed multiple large granulomatous follicles on the upper and lower palpebral conjunctiva with mucous discharge. The patient had a history of contact with a dead wild rabbit. Conjunctival cultures showed *F. tularensis* coccobacilli, and blood cultures and serological tests came back negative. The patient was admitted and treated with IM streptomycin and IV nafcillin. He had a full recovery within two months [125].

Steinemann et al. described a patient presenting with right preauricular lymphadenopathy with corneal ulceration and yellow discrete conjunctival nodules on slit lamp examination. The patient had a tick bite several days prior. Conjunctival biopsy and corneal cultures revealed giant cells suggestive of conjunctival granuloma, prompting a diagnosis of POGS. Serological testing for *F. tularensis* was strongly positive, likely due to the disease being in a later stage at time of testing [136]. She was treated with 100 mg of doxycycline hyclate twice daily for three weeks with 14 mg/mL of fortified topical gentamicin sulfate drops hourly to the right eye, but subsequently underwent a corneal transplant as medial corneal opacities persisted after resolution of the infection [137].

### 3.3.1. Atypical Ophthalmic Manifestations of Oculoglandular Tularaemia

Although most ophthalmic manifestations reported have mostly affected the anterior segment of the eye, Terrada et al. reported a case of posterior uveitis in oculoglandular tularaemia. The patient was a 52-year-old hunter who recalled having killed rabbits a few days prior. There was significant vitritis and a large yellowish lesion involving the macula with an overlying subretinal detachment, associated with subretinal haemorrhages. Optical coherence tomography scans of the macula demonstrated subretinal fluid and heaped up cells anterior to the pigment epithelium. Ultrasound biomicroscopy revealed a 3.8 mm choroidal granuloma with a few calcifications in the left eye. Anterior chamber and serum samples were taken for serological testing, yielding positive indirect immunofluorescence assay results for *F. tularensis*. The patient was treated with 200 mg of doxycycline once daily for three weeks. The vision loss persisted with a central scotoma [130].

Pärsinnen et al. described a 58-year-old female who presented with acute right raised intraocular pressures (IOP) and right corneal oedema secondary to oculoglandular tularaemia. Examination measured intraocular pressures of 68 mmHg. There was lid oedema and erythema, and conjunctival hyperaemia without discharge. The cornea was oedematous. She also had fever, myalgia, and right eye pain. Cervical and preauricular lymph nodes were assessed to be normal, and the right inguinal lymph nodes were enlarged. Two serum samples were taken 17 days apart (Day 1 and Day 18) for ELISA testing for tularaemia antibodies; the first result was negative, and the second confirmed acute tularaemia infection. Other laboratory investigations showed raised inflammatory markers and leukocytosis. The IOP was initially lowered with two 0.5% timolol eye drops instilled into the right eye and 250 mg of IV acetazolamide. Subsequently, due to a shallow anterior chamber, laser iridotomies were performed. 500 mg of ciprofloxacin was given twice daily for 10 days to treat tularaemia infection. At the two-month follow-up, there remained mild corneal oedema. By the last follow-up, the right eye still had a raised IOP at 26 mmHg [138]. The visual field by perimetric testing remained normal.

Dacryocystitis has also been reported in oculoglandular tularaemia. The patient was a 27-year-old, 18-week-pregnant female who presented with unilateral acute dacryocystitis and purulent conjunctivitis in her right eye, associated with right-sided preauricular and submandibular lymphadenopathies. The patient had no exposure to infected animals or tick bites, but it was noted that she lived in Turkey, where tularaemia is endemic. The surgical drainage of dacryocystitis was completed, and the drained contents were sent for investigation. A polymerase chain reaction (PCR) of the drained contents was weakly positive for *F. tularensis*, and Gram-negative bacteria was cultured on human blood agar. Fine needle aspiration cytology of the submandibular lymph nodes revealed suppurative inflammation—polymorphonuclear cells with no bacteria seen. Serological microagglutination testing for *F. tularensis* antibodies was positive. The patient was given a 14-day course of oral amoxicillin-clavulanic acid in view of her pregnancy and gentamicin

eye drops, which allowed resolution of the purulent conjunctivitis in one week. A second surgical drainage was executed with a ciprofloxacin-impregnated sponge inserted into the tissue defect following drainage. The patient had a full recovery without recurrence [139].

3.3.2. Ophthalmic Manifestations in Other Subtypes of Tularaemia, Other Than Oculoglandular Tularaemia

Dacryocystitis and nasolacrimal duct obstruction are possible complications in oropharyngeal tularaemia, as described by Köse et al. [140]. The patient was a 33-year-old male who presented with submandibular lymphadenopathy, fatigue, nausea, and vomiting. Serum agglutination test for *F. tularensis* antibodies was positive. He was initially treated with streptomycin and doxycycline, achieving resolution of lymphadenopathy after several months. He then developed recurrent swelling around the nasolacrimal sac associated with tearing, redness, and purulent discharge of the right eye. Slit lamp and fundoscopic examinations were unremarkable. Nasolacrimal lavage was done, and purulent discharge ensued. He was started on 1 g of oral amoxicillin-clavulanic acid twice daily before cultures for the discharge were taken. Antibiotic therapy was halted as cultures were negative. He subsequently underwent otorhinolaryngology evaluation in view of dacryocystorhinostomy surgery, which he ultimately declined. Köse et al. proposed that the mode of spread was via lymphatics from the cervical lymph nodes to lymphoid tissue near the nasolacrimal duct. Alternatively, inflammation of the nasolacrimal duct could have been due to its close proximity to the ongoing cervical lymph node inflammation. Ductal inflammation led to tissue oedema in the mucous membranes, and subsequently fibrosis in the nasolacrimal duct. The recurrent episodes of dacryocystitis could have arisen from obstruction of the fibrosed nasolacrimal duct [140].

Marcus et al. highlighted ophthalmic findings in a 25-year-old male with typhoidal tularaemia. The patient complained of blurring of vision two days post-laparotomy for an open lung biopsy, which was done as he initially presented with meningitis, hepatitis, and lung infiltrates requiring mechanical ventilation. The biopsy yielded negative results. His visual acuity then was 20/200 in the right eye and 20/30 in the left eye. Fundoscopy revealed deep retinal haemorrhages, superficial retinal infiltrates, roth spots and a subfoveal inflammatory lesion with radiating striae in the right eye, whereas the left eye showed a superficial retinal infiltrate. An increase in the serum agglutination titre for *F. tularensis* antibodies was diagnostic for typhoidal tularaemia. He was treated with gentamicin sulfate and tetracycline hydrochloride. The outcome was complete systemic recovery, with complete restoration to visual acuity of 20/20 bilaterally and normal fundal appearance [141].

### 3.4. Babesiosis

*Babesia microti* is a protozoan parasite transmitted by the *Ixodes* tick, mainly causing babesiosis in the Northwest and upper Midwest regions of the U.S.A. Both adult and nymphal ticks transmit the pathogen, with the latter being the commoner vector from late spring to summer. *Babesia divergens* is the main aetiological organism for babesiosis in Europe [5,41].

Although rarely involving the eye, ocular presentations of babesiosis include surface manifestations such as conjunctival injection, and posterior findings include retinal haemorrhages [41]. Zweifach et al. also reported retinal nerve fibre layer infarct in babesiosis infection; a 52-year-old male had presented with a scotoma in his right eye with a background of 10-day history of anaemia and fever. He had visited Shelter Island, in Long Island, New York, two months prior. Ophthalmic examination revealed a large nerve fibre

layer infarct superotemporal to the optic disc in the right eye, and a similar infarct at the inferior edge of the left disc. Babesiosis infection was confirmed by a peripheral blood smear that showed intra-erythrocytic parasites consistent with *Babesia microti*. A full blood count revealed anaemia and low haematocrit. The patient was stabilised medically and subsequently had resolution of both infarcts. However, his visual complaint persisted. Zweifach et al. postulate that infected erythrocytes confer prothrombotic effects, resulting in these infarcts [142]. Infected erythrocytes may be sequestered in small vessels via interactions between the erythrocyte and vascular endothelial receptors, resulting in micro-thrombus formation. Alternatively, the parasitised erythrocytes may trigger an acute inflammatory response and activation of the coagulation cascade, causing increased erythrocytes to adhere to retinal capillary endothelium and thrombi to form [41].

Conjunctivitis has also been reported in babesiosis. Strizova et al. reported a case of a 36-year-old male with a history of a road traffic accident requiring multiple blood transfusions who presented with eyelid oedema, conjunctivitis, arthritis, and urethritis, the latter three mimicking the classic triad of Reiter's syndrome. The patient denied any history of tick bite. Detailed examination by an ophthalmologist and conjunctival swab revealed no abnormalities; rheumatological investigations also ruled out Reiter's syndrome. The patient then developed eyelid swelling which prompted an extensive investigation for multiple pathogens. Subsequently, babesiosis infection was confirmed by lymphocyte transformation test (LTT). He received proper antibiotic therapy for 28 days consisting of doxycycline (200 mg/day) and trimethoprim-sulfamethoxazole (160/180/per day), achieving complete resolution four weeks after initiation of treatment [143]. Fitzpatrick et al. also reported conjunctivitis in a 47-year-old male patient with a history of splenectomy and recent intraoperative blood transfusion [144] who was later found to have babesiosis.

### 3.5. Tick-Borne Relapsing Fever

Tick-Borne relapsing fever is a spirochaetal disease with multiple causative organisms from the *Borrelia* genus. Of note, *Borrelia hermsii* and *Borrelia turicatae* are responsible for most cases in the United States [5]. Ticks of the *Ornithodoros* genus (e.g., *Ornithodoros hermsi* Wheeler, Herms & Meyer) serve as the primary vector in spreading these spirochaetes. The disease is endemic to the western United States, plateau regions of Mexico, Central and South America, the Mediterranean, Central Asia, and much of Africa [145].

In a case series reviewing 182 patients with tick-borne relapsing fever (133 confirmed and 49 probable cases) in the northwestern United States and southwestern Canada, of the number of patients for whom information was reported, 25% (12/48) had photophobia, 26.3% (10/38) had eye pain, and 4.5% (2/44) had conjunctival injection [146]. Ocular manifestations of the disease include anterior uveitis, intermediate uveitis, optic neuritis [147], and choroiditis [148].

### Ocular Inflammation

Lim and Rosenbaum reported a case of a 12-year-old boy who had a prodrome of flu-like symptoms, headaches, rash, and formed visual hallucinations following a camping trip in eastern Oregon [149]. Lyme disease serology immunoglobulin M (IgM) was elevated despite the lack of exposure to endemic areas of *B. burgdorferi*, and is likely explained by serologic cross-reactivity which occurs within *Borrelia* species [150]. *B. hermsii* IgM and immunoglobulin G (IgG) were elevated as well. Ceftriaxone followed by cefuroxime was initiated for the patient but he soon noted floaters and blurred vision in the right eye (best-corrected visual acuity was 20/40 in the right eye and 20/20 in the left eye). Fine keratic precipitates, anterior chamber cells and flare, and anterior vitreous cells were noted on examination of the right eye. Topical prednisolone acetate did not result in improvement,

but rapid resolution of the inflammation with oral doxycycline 100 mg/day for four weeks and the timing of the ocular presentation suggests that the patient's vitritis was metastatic endophthalmitis from the seeding of *B. hermsii*.

Hamilton also described three cases presenting with acute iridocyclitis and one patient with chronic cyclitis accompanied by persistent headache [147]. All these patients had gross vitreous exudates, and visual acuities ranging from 6/9 to <6/60 in the affected eye. The conjunctivae of the three patients with acute iridocyclitis was injected. As the condition of iridocyclitis often comes after the last relapse of fever, it is challenging to recover the spirochaete from the patient's serum. In two of the patients, the spirochaete was never found. Patients with posterior synechiae were given mydriatic agents (atropine), 2% cocaine eye drops, and heat. One patient developed marginal keratitis followed by multiple corneal erosions and was treated with 2% silver nitrate. All four patients recovered with excellent prognosis.

### 3.6. Ehrlichiosis

Human monocytic ehrlichiosis (HME) is caused by *Ehrlichia chaffeensis*, which can be carried by vectors like *A. americanum* (Lone Star tick) and *D. variabilis* (dog tick) and animal hosts such as the white-tailed deer, dogs, and foxes. PCR has also been used to detect DNA of *E. chaffeensis* in other tick species, particularly *I. pacificus*, *I. ricinus*, *A. testudinarium*, and *Haemaphysalis yeni* [151]. HME mainly affects monocytes. Constitutional symptoms of ehrlichiosis include fever, headache, rigors, malaise, and myalgia [152].

Ocular manifestations are highly uncommon in HME. Those previously reported include ophthalmoplegias such as fourth cranial nerve palsy, and optic neuritis [153,154]. Orbital myositis was described in a patient who was presumed to have co-infection of Lyme disease, HME, and babesiosis [155]. Raja et al. reported a case of posterior uveitis likely caused by ehrlichiosis. The patient was a 68-year-old female who had a four-year history of bilateral decrease in visual acuity, with a purported worsening over the past month. Her social history uncovered numerous recent tick bites. Fundoscopy showed presence of epiretinal membranes and cryopexy scars in both eyes. There was cystoid macular oedema in both eyes, and fluorescein angiogram showed retinal vasculitis. Biochemically, a tick-borne infection panel was done and showed elevated *E. chaffeensis* titres. The patient was started on a prophylactic course of 100 mg oral doxycycline and 40 mg sub-Tenon triamcinolone for both eyes. The patient had a reduction in cystic macular oedema [156].

### 3.7. Rocky Mountain Spotted Fever

Rocky Mountain Spotted Fever (RMSF) is the most frequent cause of fatality from tick-borne disease in the United States [157]. Caused by the intracellular bacterium *Rickettsia rickettsii*, RMSF is transmitted by a dog tick (*D. variabilis*) in the eastern two-thirds of the United States, whereas the Rocky Mountain wood tick (*D. andersoni*) is the main vector in the Western states [158]. Locations where RMSF has been previously thought to be rare or nonexistent have now been affected; in Arizona, where the *Dermacentor* species ticks are rare, the brown dog tick *Rhipicephalus sanguineus* s.l. Latreille has been implicated in the transmission of RMSF [159]. In the United States, cases peak seasonally from May to August every year [160]. It is said that although the disease can occur throughout the year, RMSF cases mainly peak during the summer and spring seasons when the ticks are the most active [161] and outdoor activity is most frequent. However, cases have also been reported to occur during winter, especially in the Southern United States [162].

The *R. rickettsii* intracellular bacterium multiplies in vascular endothelial cells of small to medium-sized blood vessels, leading to vasculitis, cell-to-cell spread of the bacterium, and subsequent cellular and tissue necrosis [158].

The disease is characterized by the classic clinical triad of headache, fever, and petechial rash, with other symptoms including myalgias, arthralgias, abdominal pain, nausea, and vomiting at presentation. Severe cases may lead to complications such as renal failure, hepatomegaly, meningismus, and neurologic abnormalities [41].

Ocular manifestations are common and involve vasculitis of the conjunctiva, retina, and choroid. Retinovascular changes include cotton wool spots, macular oedema, vascular sheathing, venous tortuosity, retinal artery or vein occlusion, and intraretinal haemorrhage and exudates. Neuro-ophthalmic manifestations have also been described—optic disc oedema owing to ischemia and inflammation, optic neuritis, optic neuropathy, and neuroretinitis, whereas anterior uveitis has been less commonly reported [41]. Ocular changes usually resolve after a few weeks of therapy [163].

### 3.7.1. Retinovascular Changes

Retinovascular changes in six patients visible on fundoscopy have been described by Presley. All six patients had positive OX-19 reactions (Weil-Felix test), and four of them revealed a positive tick bite history. The retinovascular changes reported include engorgement and tortuosity of the retinal veins, peripapillary and posterior pole oedema of the retina, bilateral papilloedema, cotton-wool exudates, flame-shaped haemorrhages, retinal haemorrhages at the posterior pole resembling Roth spots, and occlusion of a branch of the inferior temporal artery [164].

Similarly, Smith and Burton reported retinal manifestations in a nine-year-old girl with a known history of a tick bite who presented with constitutional symptoms. This was followed by a rash with irregularly defined macules on her arms and legs which spread to the trunk. Fundoscopy revealed bilateral optic disc oedema, cotton wool spots and tortuous and dilated retinal veins. Fluorescein angiography showed focal areas of capillary nonperfusion with blocked retinal and choroidal fluorescence at the cotton wool spots. Perivascular staining adjacent to the infarcted areas, hyperfluorescence of the optic discs, and prolonged retention of intravascular fluorescein suggesting venous outflow obstruction were demonstrated as well. The patient was previously treated with systemic cephalexin monohydrate and amoxicillin without improvement. A complement fixation test for RMSF was positive. The patient responded to IV chloramphenicol, and was discharged on oral tetracycline. It is suggested that following necrosis of vascular endothelial cells, the necrosis of the intima and media causes retinal arteriole thrombosis, resulting in areas of microinfarcts and cotton wool spots. There is also occlusion of small vessels within the optic nerve head, producing endothelial incontinence, leakage, and subsequent optic disc oedema. The disc oedema causes partial obstruction of the central retinal vein, causing vein dilatation and tortuosity [165].

A case of paediatric retinitis was described by Moore et. al. in a 14-year-old male who presented with an acute onset of systemic febrile illness associated with bilateral conjunctival injection and periorbital oedema. The patient recently returned from travel to Mexico and denied any history of tick bites. Due to his rapid deterioration into septic shock, the patient was admitted to the paediatric ICU. He was intubated and started on broad-spectrum antibiotics, including doxycycline. Infectious workup demonstrated positive rickettsial IgM serology but negative IgG serology, with other viral and bacterial workup unremarkable. The patient improved with systemic antibiotics but subsequently complained of his right eye blurring more than his left. Best-corrected visual acuity was measured at 20/30 + 2 in the right and 20/20 in the left eye respectively. On slit lamp examination, mild trichiasis, a right subconjunctival haemorrhage, and bilateral punctate epithelial erosions were noted. Discrete raised white lesions were also found bilaterally on

dilated fundus examination, with scant dot-blot haemorrhages noted in the left eye. This was consistent with rickettsial retinitis. The patient's vision loss was attributed to surface disease secondary to trichiasis and corneal irritation, which resolved with artificial tears. He was discharged home after completion of 14 days doxycycline and follow-up at the eye clinic showed visual acuity improvement to 20/20 in both eyes with interval improvement of his bilateral retinitis. Fundus photos also demonstrated resolving white retinal lesions [166].

### 3.7.2. Neuro-Ophthalmic Manifestations

Nazarian et. al. reported bilateral anterior optic neuropathies after an erythematous centrifugal rash in a 58-year-old farmer with a known tick bite history who presents with a two-day history of sudden, painless binocular vision loss. Visual acuity declined to 20/150 in the right eye and 20/200 in the left eye, with reduced bilateral colour vision. His pupils reacted sluggishly to light without a relative afferent pupillary defect. Fundoscopy showed swollen bilateral optic discs and dilated retinal veins. Investigations included a lumbar puncture, which revealed a normal opening pressure, elevated protein, red blood cells, and white blood cells (90% lymphocytes). RMSF IgG was positive by solid-phase enzyme immunoassay, which was confirmed by indirect immunofluorescence assay, thereby supporting the diagnosis of RMSF. The patient had a gradual recovery after being treated with 100 mg oral doxycycline every 12 h for 60 days [167].

Neuroretinitis with bilateral macular stars were also described in an 86-year-old female who developed bilateral subnormal visual acuity (20/40 in the right eye and counting fingers vision in the left eye) without fever or skin rash. Cerebrospinal fluid analysis showed lymphocytic pleocytosis and elevated protein, and the markedly positive serum *R. rickettsii* antibody test helped establish the diagnosis of RMSF. This was the first case of neuroretinitis from RMSF that did not have associated constitutional symptoms and had a macular star appearance [168].

### 3.7.3. Uveitis

Finally, although uncommon, the first case of anterior uveitis in association with RMSF was reported by Cherubini and Spaeth in a 54-year-old female who presented with headache, cough, fever, dysuria, maculopapular skin lesions, and blurred vision. Examination revealed corrected visual acuity of 6/22 in the right eye and 6/15 in the left eye. There was right conjunctival injection with miosis of the pupil that was almost non-reactive to light, while the left pupil was normal in size but reacted sluggishly to light and accommodation. The right cornea had many small white keratic precipitates spread diffusely through the entire endothelium with several large yellow keratic precipitates centrally, and the right eye aqueous contained many cells with a mild flare. This is compared to the left eye which had a few small white keratic precipitates, with the aqueous showing a mild cellular reaction. Investigations revealed raised total white blood cells, a negative Weil-Felix reaction and a slight increase in beta-2 globulin fraction on protein electrophoresis. Further investigations included a punch biopsy of the skin of the right leg which demonstrated histopathologic features compatible with RMSF. Treatment with IV tetracycline and supportive therapy led the patient to uninterrupted recovery, and toward the end of her hospital stay, the Weil-Felix test became positive for OX-19 reaction in a dilution of 1/320 and a specific complement fixation test was positive for RMSF [169].

### 3.8. Mediterranean Spotted Fever

Similar to Rocky Mountain Spotted Fever, Mediterranean Spotted Fever belongs to the spotted fever group of *Rickettsia*. The causative agent is *Rickettsia conorii*, an obligate intracellular Gram-negative bacilli that is transmitted by the dog tick *R. sanguineus* s.l. [170].

The disease is endemic to the Mediterranean area including Southern Europe and Northern Africa, as well as to South Africa and areas surrounding the Black Sea [171]. The pathophysiology of the disease is similar to that of Rocky Mountain Spotted Fever, and involves the invasion of *R. conorii* into endothelial cells lining small and medium-sized blood vessels [172].

Ocular manifestations of Mediterranean Spotted Fever include that of multifocal retinitis [173,174], Parinaud's oculoglandular syndrome [175–177], corneal ulcers [176–178], ischemic optic neuropathy [179], conjunctivitis and dacryoadenitis [180], posterior segment changes [174] such as retinal vasculitis [178,181], retinal hemorrhages, branch retinal artery and vein occlusions, retinal detachment, optic disc oedema [182,183], cotton-wool spots [181], and retinitis with mild vitreous inflammatory reaction [183].

### 3.8.1. Parinaud Oculoglandular Syndrome in Mediterranean Spotted Fever

POGS can occur in patients with Mediterranean Spotted Fever, as described by Abroug et al. The patient was a 66-year-old male who presented with a month-long history of left-sided unilateral persistent granulomatous conjunctivitis, associated with left preauricular lymphadenopathy. He recalled an incident where a jet of contaminated water projected into his eye. He presented with grossly swollen eyelids, conjunctival hyperaemia, and chemosis with mucopurulent discharge. Serological indirect immunofluorescence assay yielded a positive result for *R. conorii* antibodies. On diagnosis of POGS secondary to Mediterranean Spotted Fever, he was treated with a two-week course of 100 mg oral doxycycline twice daily and achieved full recovery [175]. As mentioned earlier, POGS may be present in oculoglandular tularaemia infections. Chmielewski et al. reported the presence of *Rickettsia* species and *F. tularensis* co-infection, which is important for appropriate antibiotic therapy for treatment success [184].

### 3.8.2. Corneal Manifestations

Alio et al. documented a case of a 69-year-old male with keratitis secondary to *R. conorii* infection. Prior to his ocular complaints, he had an 11-day history of constitutional symptoms associated with a tick bite on his left leg. The diagnosis of Mediterranean Spotted Fever was confirmed by a positive serological testing for *R. conorii* antibodies. He then complained of left eye pain, tearing, and photophobia, prompting an eye examination which showed an ameboid-type corneal ulcer and ciliary injection. A slit lamp examination demonstrated corneal oedema, corneal inflammatory infiltrates, and mild infiltration of the anterior stroma, while fundoscopy showed retinal vasculitis. Corneal scrapings and tear samples were also collected and sent off for immunofluorescence testing (direct and indirect immunofluorescence respectively) and returned positive, suggesting the presence of active rickettsiae in the corneal ulcer. The patient was treated with oral doxycycline, 1% atropine with tetracycline ointment every 12 h and occlusion. The ulcer completely resolved by the third day of topical treatment [178].

### 3.8.3. Posterior Segment Manifestations

Unilateral acute anterior ischemic optic neuropathy associated with Mediterranean Spotted Fever has been reported by Khairallah et al. in a 43-year-old male who first presented with constitutional symptoms and a maculopapular skin rash without a known history of a tick bite. Following the onset of fever, he noticed painless acute visual loss in the right eye with a best-corrected visual acuity of counting fingers in the affected eye, associated with relative afferent pupillary defect. Fundoscopy of the right eye revealed optic disc oedema, optic disc and retinal haemorrhages, and small white retinal lesions in the periphery and posterior pole. A superior altitudinal defect in the affected eye was also

detected on visual field testing. The patient tested positive for *R. conorii* through the indirect immunofluorescence test, and was then treated with 200 mg/day doxycycline for a week, which resulted in improved visual outcomes. It is suggested that *R. conorii* may have invaded the optic nerve head microvessels, causing endothelial injury and tissue necrosis with subsequent optic disc infarction, or the ischemic event might have resulted from an immune-mediated inflammation [179].

Khairallah et al. also investigated other posterior segment manifestations in 30 patients with Mediterranean Spotted Fever. All patients tested positive for the serologic detection of antibodies to *R. conorii* by indirect immunofluorescence at the acute stage of the disease, supporting the diagnosis of Mediterranean Spotted Fever in all patients. All 30 patients (60 eyes) underwent a complete ophthalmic evaluation consisting of best-corrected visual acuity, slit lamp examination, tonometry, dilated biomicroscopic fundus examination with contact lenses, fundus photography, and fluorescein angiography. Of the 30 patients, 25 patients had unilateral (n = 5) or bilateral (n = 20) posterior segment involvement. A total of 64% of the 25 affected patients had ocular symptoms, and 36% of them had ocular complaints including decreased vision, paracentral scotoma, floaters, and ocular redness due to conjunctivitis or anterior uveitis. The best-corrected visual acuity at presentation ranged from 20/100 to 20/20. Mild vitreous inflammation was present in 45 eyes (75%), and other fundus and fluorescein angiographic changes were found: juxtavascular white retinal lesions, focal vascular sheathing, multiple arterial plaques, haemorrhages (intraretinal, white-centred retinal, subretinal), serous retinal detachment, macular star, cystoid macular oedema, optic disc oedema, branch retinal artery occlusion, optic disc staining, retinal vascular leakage, delayed filling in a branch retinal vein, and multiple hypofluorescent choroidal dots [174].

All patients were treated with a two-week course of oral doxycycline, initiated before ophthalmic examination in 25 patients and after ophthalmic examination in five patients. The posterior segment changes had a self-limited evolution in 24 of 25 affected patients, with a final visual acuity of 20/20 in 42 out of 45 affected eyes. Retinal neovascularization was detected within the area of retinal pigment epithelium changes in one eye at six months after initial presentation, but no further abnormalities were reported over a follow-up duration of six months in this affected eye [174].

*3.9. Toxoplasmosis*

Toxoplasmosis is the most common cause of infectious posterior uveitis [185]. It is caused by *Toxoplasma gondii*, an obligate intracellular protozoan, which exists in three forms: the trophozoite, cyst, and oocyst. Transmission of *T. gondii* occurs between definitive and intermediate hosts; definitive hosts are members of the family Felidae while intermediate hosts are warm-blooded animals (mammals and birds), including humans [6]. Transmission to humans is commonly known to be through the ingestion of cysts in contaminated meat (especially pork and lamb), through consuming sporulated oocysts from water, soil, or food contaminated indirectly from cat faeces, direct contact with contaminated cat faeces, or through transplacental transmission of trophozoites by infected pregnant mothers [6,186].

However, oral transmission alone may not explain the widespread distribution of *T. gondii* over a wide variety of host animals and geographic areas. The possibility of *T. gondii* being an unrecognised tick-borne pathogen helps to explain the prevalence of the parasite in herbivorous mammals that are unlikely to contract primary infection orally with meat or cat faeces, as well as in other hosts that do not possess common risk factors such as cat ownership. Hence, studies have suggested the possibility of transmission of

*T. gondii* in a number of ticks, including *D. variabilis*, *D. andersoni*, *A. americanum*, *D. reticulatus* (Fabricius), *Ixodes ricinus* (Linnaeus), *Amblyomma cajennense* (Fabricius) complex, mainly *Amblyomma sculptum* Berlese, *Ornithodoros moubata* (Murray), and *H. longicornis* Neumann [6].

### 3.9.1. Posterior Uveitis

Ocular manifestation of toxoplasmosis is mainly posterior uveitis, particularly focal retinochoroiditis. In a study evaluating the distribution of different uveitis entities and their associations with infections in 160 consecutive uveitis patients, a direct infection was suggested to be linked with uveitis in 23 patients (14.4%). In the 23 patients, Lyme borreliosis and toxoplasmosis were the most frequently seen, in seven patients each (4.3% out of all 160 patients). Six patients with toxoplasmosis had posterior uveitis, and one patient had panuveitis [187].

Toxoplasmic retinochoroiditis is typically unilateral. Patients may initially present with floaters and decreased visual acuity, with a dense vitreous reaction which forms a characteristic "headlight in the fog" appearance [186]. Active lesions are usually seen as whitish foci and are frequently adjacent to a pigmented or atrophic scar. With resolution of the disease over weeks to months, the active lesion will eventually be replaced by an atrophic retinochoroidal scar [188].

Less common presentations include associated serous macular detachment [189], retinal vasculitis, neuroretinitis [190], papillitis [191], disc haemorrhages with venous engorgement, and macular star [184]. Other complications of ocular toxoplasmosis include chronic iridocyclitis, cataract formation, secondary glaucoma, band keratopathy, cystoid macular oedema, retinal detachment, and optic atrophy secondary to optic nerve involvement [188].

Diagnosis is primarily clinical and anti-toxoplasma IgM and IgG can support the diagnosis. However, these antibodies appear early and disappear rapidly in acute infection and are thus hard to detect in patients. The Sabin-Feldman dye test has been the gold standard serologic test for toxoplasmosis, but is being replaced by enzyme-linked immunosorbent assays for anti-toxoplasma antibodies [186]. Treatment is typically recommended for lesions involving the macula, peripapillary region, in proximity with a large retinal vessel, severe vitritis, or in immunocompromised patients. This typically involves a combination of antiprotozoal agents. Systemic steroids may be added if there are no contraindications. Topical steroids and cycloplegics can be used for anterior segment inflammation if present.

### 3.9.2. Primary Intraocular Lymphoma

Although there is no evidence for direct causation, an association between primary intraocular lymphoma (PIOL) and toxoplasmosis has been reported. A 79-year-old female with previous cataract surgeries to both eyes was found to have cells and flare of the anterior chamber and vitreous cells. On examination, the right eye showed an area of focal retinitis with surrounding pigmentary changes of the superior equatorial retina and macular pigmentary changes. She was treated with presumptive ocular toxoplasmosis given her elevated serum toxoplasmosis IgG titres. Initial neuroimaging for lymphoma revealed no lesions. However, she experienced two episodes of presumed recurrence and a diagnostic vitrectomy was subsequently performed in view of progressive vitritis, with a diagnosis of large B cell lymphoma being formed. Further neuroimaging revealed central nervous system lesions. The patient was started on systemic chemotherapy [192].

### *3.10. Powassan Encephalitis*

Powassan encephalitis is caused by the Powassan virus, which can be transmitted by various tick species, including *Ixodes cookei* Packard, *Ixodes marxi* Banks, *Ixodes spinipalpis* Hadwen and Nuttall, *I. scapularis*, and *D. andersoni*, usually in the summer months [5]. The

Powassan virus is an RNA virus of the Flaviviridae family with rare but rising incidence, with cases concentrated in the Northern Great Lakes regions of the United States [193,194]. Between 2011 and 2020, a total of 194 Powassan virus-related neuroinvasive and non-neuroinvasive infections have been reported by the United States Centers for Disease Control and Prevention (U.S. CDC) [195].

Ophthalmic manifestations previously reported include retinal vein tortuosity and optic disc oedema [41]. Lessell et al. delineated a case of a 53-year-old who complained of diplopia and was found to have near-total ophthalmoplegia, with accompanying mood changes, memory impairment, and neurological symptoms. Serological investigations showed positive Powassan-specific IgM and multiple areas of T2 hyperintensity in the cerebral hemispheres and cerebellum on brain MRI. The patient was treated with ceftriaxone, ampicillin, and acyclovir, with reported resolution of her systemic symptoms. Although optic nerve function remained intact and structural examination of the globe was normal, loss of saccades, an inability to maintain eccentric gaze, and impaired smooth pursuits remained. Doll's eye movements were preserved, although conjugate excursions were slowed and delayed. Further improvements in the ability to initiate up-gaze was noted after three months [196].

Nord et al. also described a patient with Powassan encephalitis presenting with a similar ophthalmic manifestation as that of the West Nile virus, in which both are flaviviruses. The patient was a 51-year-old male with a seven-year history of chronic bilateral non-infectious anterior and intermittent uveitis with secondary macular oedema who had been treated to disease quiescence. He developed fever and altered mental status following a tick bite, and an encephalitis workup was performed. Serological results for West Nile virus and Lyme disease were negative. Lumbar puncture demonstrated Powassan virus-specific IgM and neutralising IgG antibodies. He was treated and recovered fully from the infection. Subsequently on follow-up, there was new disease activity with increased anterior chamber and vitreous cells; fundoscopy showed new multifocal choroidal lesions in linear streak-like distributions. Fundus autofluorescence delineated hypo-autofluorescent streaks while fluorescein angiography revealed extensive multifocal choroiditis. The authors justified that bilateral multifocal chorioretinitis with linear streaking was a typical ophthalmic finding for West Nile virus infections, and that such chorioretinitis is the result of simultaneous viral-mediated direct damage and tissue inflammatory response to the virus. They postulated that the Powassan virus had spread via the central nervous system to the choroid, and less likely hematogenously through the choroid to the retina [193].

### 3.11. Tick-Borne Encephalitis

Hallmarked by neurological manifestations following a viral prodrome, only a single case report exists regarding ocular manifestations with tick-borne encephalitis (TBE). Voulgari et al. reported a case of a 58-year-old male patient with no significant medical history who presented with five days of occipital headaches, poor balance and a history of a tick bite six weeks prior. He developed unilateral right eye vision loss three days later with examination revealing non-granulomatous anterior uveitis, vitritis, and retinal haemorrhages. Optical coherence tomography and fundus fluorescein angiography-indocyanine green angiography (FFA-ICG) were performed and excluded macular oedema and papillitis. Seroconversion of TBE virus-immunoglobulin titres was observed two weeks later with the authors concluding TBE-related uveitis. The patient was treated with topical prednisolone

acetate 1% six times a day. He also received empiric intravenous acyclovir 10 mg/kg three times a day to cover for possible herpes meningoencephalitis with retinal involvement, although PCR of the cerebrospinal fluid returned negative for all Herpes family viruses and the acyclovir was subsequently discontinued. Supportive care was pursued, and the neurological symptoms resolved completely in three weeks along with complete resolution of intraocular inflammation without sequelae at five weeks [197].

### 3.12. Colorado Tick Fever

Colorado tick fever is caused by the Colorado tick virus, an RNA virus in the genus *Coltivirus*, which is transmitted by the primary vector *D. andersoni*, a wood tick. The disease is endemic to elevations of 4000 to 10,000 feet in the Rocky Mountain region of the United States and southwest Canada [41] (provinces of Alberta and British Columbia) [5]. Patients may present with fever of a biphasic pattern; symptoms remit after two to four days but recur one to three days later [198]. Associated symptoms include that of chills, headache, retro-orbital pain, photophobia, myalgias, generalised weakness, pharyngeal erythema, and maculopapular or petechial rash [198]. Ocular manifestations are limited to retro-orbital pain, photophobia, and conjunctival injection [5]. Life-threatening complications, such as meningoencephalitis and those associated with disseminated intravascular coagulation, are rare and occur more commonly in children [197]. No specific antiviral therapies are available, and supportive care is the mainstay of treatment for this disease [199].

Appendix A (Table A1) shows the various ocular manifestations for the tick-borne diseases covered in this review, as well as the rendered treatment and outcomes.

## 4. Discussion

This article seeks to provide a comprehensive review of the various reported ocular manifestations of tick-borne diseases, as well as provide suggestions for tick removal and subsequent treatment, for which there are currently no clear guidelines. In addition, the relevance of climate change and air travel is highlighted in this article, with the aim to increase awareness of tick-borne illnesses especially in countries less familiar with such diseases.

### 4.1. Proposal for Removal of Ticks from the Eye Surface and Subsequent Management

In this discussion, we would propose a standard procedure for safe removal of ticks from adnexal structures.

After the identification of the tick, the focus should be on complete removal of the tick, as far as possible. Lai et al. describes a process in which the surface is cleansed with topical chlorhexidine 0.05% [8]. This reduces the risk of contact dermatitis [200] and secondary infection of the resulting wound from skin commensals. Patient comfort during the procedure should be accounted for with either application of topical anaesthesia (topical tetracaine 0.5%) or a local infiltration of anaesthesia [12] for more deeply embedded ticks.

We suggest the use of toothed forceps in the removal of the tick for better grasp of the body [8]. Blunt, curved, medium-point forceps may be used safely and effectively to dislodge the tick [10]. Removal should be gentle with the goal of not crushing the body and removing the tick as completely as possible, especially the hypostome. Clinicians should grasp the hypostome as close to the skin as possible during removal, with steady traction applied directly away from the skin. Forceful removal may result in the release of toxins from the tick or retained parts [9,20,201]. Components of the tick left behind are known to lead to abscess or granuloma formation; hence a recommendation by Park et al. for complete removal with punch biopsy [20], with the drawback being invasiveness and less satisfactory cosmesis. We would like to propose examination of the removed tick with

consideration for incision and exploration should there be suspicion of retained elements. The Infectious Diseases Society of America 2020 guidelines recommend that any remnants of a partially removed tick that poses a risk of Lyme disease be left alone and permitted to fall out, with more value given to the removal of the tick within 48 h. However, the guidelines also acknowledge that other tick-borne pathogens may require less attachment time to infect a host [202]. Hence, in view of potential infection by other pathogens and abscess or granuloma formation, we would like to propose examination of the removed tick and consideration be made for further exploration in the case of incomplete removal on an individual case basis.

Attention to tick collaterals (e.g., eggs) should be had, with removal of these as well using gentle irrigation with chlorhexidine. The eyelids, should they be involved, should be everted and the fornices swept to remove any remaining debris. This may be performed with a cotton-tipped applicator.

Thereafter, we recommend the application of topical antibiotics for at least two weeks (Lai et al. chose 0.3% tobramycin ointment [8], Shrestha et al. used chloramphenicol ointment and dexamethasone sodium phosphate for two weeks [11]) and close follow up of the case.

The decision for systemic antibiotics should be based on an assessment of risk factors. We propose (based on the U.S. CDC criteria) duration of the attached tick (>36 h), identification of the tick (adult or nymphal), whether prophylaxis was administered < 72 h, endemicity, systemic presentation, and extralocal manifestations outside the tick-bite area [203,204]. Further workup should be pursued based on history (particularly travel to endemic areas) and systemic manifestations.

*4.2. Prototype: Lyme Disease, The Immune System, and Pathways for Tick-Borne Disease Entry into the Eye*

In our review, we noted more extensive literature was available for Lyme disease. We found that tenets for Lyme disease recognition and management can serve as a prototype for the approach to other tick-borne illnesses.

The tenet of Lyme disease treatment is that of early recognition and the administration of systemic antibiotics [88,205].

Early recognition is crucial, for delayed disease and its associated complications may result, as described in the stages of Lyme disease above. Prophylaxis for Lyme disease after a high risk tick bite, as stated in the Infectious Diseases Society of America (IDSA) guidelines for adults is a 200 mg single dose of PO doxycycline [202].

The current recommendations by the U.S. CDC are as follows:

- Erythema migrans: PO Doxycycline 100 mg BD (10–14 days), or PO Amoxicillin 500 mg TDS (14 days), or PO Cefuroxime 500 mg BD (14 days) for adults [206].
- Neurologic Lyme disease: PO Doxycycline 100 mg BD (14 to 21 days), or IV Ceftriaxone 2 g 24H (14 to 21 days) [207].
  - In known Lyme disease associated CN VII palsy, glucocorticoids are not indicated and may result in worse outcomes [94,208–210].

As mentioned, the decision for systemic antibiotics and prophylaxis for Lyme disease should be based on assessment of risk factors.

4.2.1. Entry of Borrelia into the Eye

With regard to the entry of *Borrelia* species into the eye, murine models shed some light on the pathophysiology [53,70]. *Borrelia* spirochaetes are able to pass through the blood-brain barrier in the early stages of infection, resulting in disseminated disease in murine models [211]. This early entry into the nervous system is supported by neuro-ophthalmic manifestations of Lyme disease tending to occur in the early stages.

Thereafter, being an immune-privileged site, the eye may sequester the bacteria, resulting in local manifestations of disease without systemic manifestations [62]. This is supported by Schubert et al. who demonstrated spirochaetes in the vitreous specimen of a patient presenting with unilateral vitritis and choroiditis despite the patient being seronegative for Lyme disease [62]. Intravenous ceftriaxone has good intravitreal penetration as evidenced by response in 12 patients with vitritis [212]. Furthermore, late-phase presentation of ocular Lyme disease [34], after a period of latency, suggests immunological tolerance of the eye [213].

Anterior chamber-associated immune deviation (ACAID) seems to be the most probable explanation for the protective response against inflammation in the eye. Catalysed by the introduction of antigens into the anterior chamber, ACAID is maintained through the activity of CD4+, CD8+, and CD25+ T helper cells. Cytokines, including TGF-ß, are involved as well. It is possible that the spirochaetes enter the anterior chamber via the bloodstream in the early stages of the disease via the ciliary artery anterior to the anterior chamber. Other cells involved in regulation of immune tolerance include the ciliary body pigment epithelial cells and the retinal pigment epithelium which inhibit the activation of Th1 cells via cytokines [214,215].

Immune privilege not only involves sequestration, but also active inhibition of innate and adaptive immune responses in the eye, which affects systemic response to antigens released from the eye [216]. Outside the eye, retinal antigens found in the thymus influences the T-cell response unique to the eye. However, elimination of self-reactive T cells through the thymus has been found to be incomplete. The resultant T cells that escape are tolerated systemically in healthy tissues, but with the eye being sheltered from the rest of the immune system, this peripheral tolerance may not extend to the eye. This results in the eye being susceptible to autoimmune inflammatory reactions. This suggests that late ocular manifestations of Lyme disease may be autoimmune in nature.

In fact, further translational research suggests that Lyme-induced nervous system insults include direct cytotoxicity, neurotoxic mediators, or autoimmune response from or even independent of molecular mimicry [217–219]. *Borrelia*, in murine and primate models appears to demonstrate affinity for nerve roots, leptomeninges, and dorsal root ganglia [105]. Although diagnostic lumbar punctures are not routinely performed, spirochaetes have been isolated from the cerebrospinal fluid of patients with neuroborreliosis as well [52]. Of interest, Kuenzle et al. studied cerebrospinal fluid plasma cells in a patient with neuroborreliosis and, through clonal expansion techniques, discovered that in addition to reactivity against *B. burgdorferi*, evidence suggests the induction of distinct immune responses specific to the pathogen and to self-antigens that are independent of molecular mimicry [220].

### 4.2.2. Implications for Other Tick-Borne Bacteria

In a study of cataract surgery patients by Chmielewski et al., *Bartonella* species DNA was found in intraoperative specimens of 1.8% of the 109 patients studied. Serologies of 34.8% and 4.6% of the patients demonstrated *B. burgdorferi sensu lato* and *Bartonella* species [221]. None of the patients had ocular inflammation. The presence of such findings suggest the possibility of a shared mechanism of entry for tick-borne diseases into the eye. Of note, the presence of DNA does not equate to active infection or inflammation of the ocular tissues, and all patients studied displayed no signs of ocular inflammation. Furthermore, PCR was used during analysis. PCR for Lyme disease has not been validated in ocular fluids [222,223], and there is a potential for overdiagnosis [223]. However, the presence of such DNA in the eye to begin with should be considered.

In the same vein, appropriate antibiotic selection for treatment of other tick-borne illnesses apart from Lyme should be based on suspicion of systemic involvement and risk factors of the patient.

### 4.2.3. Corollary: Tick-Borne Viruses and Entry into the Eye

Two tick-borne viruses presented in this article—the TBE virus and the Powassan virus—are both members of the Flaviviridae family of which the dengue virus is also a member. Uveitic manifestations of dengue show a higher incidence of posterior uveitis compared to anterior uveitis [224]. In vivo studies suggest CD4+CD8− and CD4−CD8+ cytotoxic T cells are sensitised to dengue viral antigens. Cross-reactivity of T memory cells leads to proliferation and an inflammatory cascade that may contribute to the breakdown of the blood-aqueous barrier, allowing entry of the virus and resulting in manifestation of uveitis [225,226]. Elucidation of the specific markers is beyond the scope of this article.

### 4.3. Climate Change and Air Travel

Climate change has contributed to disease transmission [2,3,227], with the incidence of tick-borne diseases occurring in areas once thought unsuitable for tick survival [228,229]. Given that ticks spend most of their life cycle in the environment [230], it is no leap to consider the impact of climate change on their distribution and hence disease incidence. For instance, ticks favour a high humidity of >85% and can only carry out host questing at temperatures > 7 °C [231]. Particularly, questing and diapausing ticks are vulnerable to extremes of temperature and humidity [230]. Host availability and vegetation cover are also variables that significantly impact the distribution of ticks [232]. Some ticks, such as the *Ixodes* species, are particularly sensitive to environmental conditions during parts of their life cycles. While *I. ricinus* ticks in their nonparasitic phases require a minimum relative humidity of 80% to avoid dessication [229], it may also be argued that lowered summer precipitation could favour survival of *I. ricinus* ticks by causing a change in vegetation growth—Norway spruce to beech—the fallen leaves of the latter providing an ideal microclimate for these ticks [233]. Furthermore, climate change may also affect tick-host diversity. This can be attributed to alterations of host-seeking activity of the ticks, which is in turn influenced by environmental changes. For example, abnormally warm temperatures during the winter month of February in Dorset (United Kingdom) triggered questing activity among *I. ricinus* ticks, as well as movement of cattle onto tick-infested pastures, resulting in an outbreak of bovine babesiosis [234]. The literature shows a spread of the *Ixodes* tick northward [229] since noticeable climate change began [235]. Similarly, the *Dermacentor* species of ticks have also shown an increased spread in Europe. Climate change also influences the characteristics of agricultural land, contributes to the expansion of fallow land, and thereby creates an ecological habitat preferred by *Dermacentor reticulatus* [236]. Climate suitability models predict with global warming that tick species are likely to establish populations further north in areas previously thought unsuitable. While climate change can play a major role in tick distribution, other factors not accounted for by climate modelling exist that may affect the actual distribution of ticks as the climate warms. It appears that climate change, in actuality, has an indirect impact on the distribution of infected ticks. This occurs through the effect climate change has on vegetation favoured by ticks, on the length of day affecting tick development, and thus a change in seasonal activity patterns. In conjunction, climate change influences the distribution of tick hosts and human use of land such as farming which affects disease risk and hence incidence of tick-borne illnesses.

At the same time, air travel has facilitated the inadvertent spread of ticks from endemic to non-endemic areas [237]. Heightened awareness of tick-borne diseases is particularly salient in immunocompromised individuals with relevant history. Tick-borne diseases are likely under-reported in travellers for a variety of reasons including lack of reliable microbiological tests in non-endemic areas, under-recognition, and misdiagnosis [238,239]. It is suggested that tick-borne diseases be given consideration in a febrile traveller returning from rural areas during tick-transmission months. Empiric use of doxycycline in severe illnesses is justified. Travel advisories to endemic areas should provide recommendations for avoidance of tick bites.

### 4.4. Travel Medicine: When to Consider Tick-Borne Diseases

In view that tick-borne diseases are less recognised in non-endemic areas, we would like to propose certain risk factors for consideration of further workup for tick-borne diseases. That is, patients with febrile illness and characteristic rash who have recently returned from tick endemic regions, especially in a history of exposure to rural or wooded areas and/or known tick bite. Consideration should be given to how long the tick was attached and the immune status of the patient, with higher suspicion in the immunocompromised.

The following risk factors should prompt increased suspicion of tick-borne diseases in a patient:

- exposure history to the outdoors [8,13];
- pet exposure [19,20];
- recent travel to endemic areas [11]; and
- systemic features associated with tick-borne illness, e.g., fever, characteristic rash.

The U.S. CDC recommends a two-step test for confirming Lyme disease by using serum samples for ELISA and confirmatory testing by Western blot [240].

Of relevance to ophthalmologists, in deciding whether to work up for Lyme disease, routine testing in patients without suggestive history is not recommended, especially in uveitis patients where the test appears to be overperformed even in endemic areas [241–245]. Conversely, in cases where seronegativity is present, but there is high suspicion of infection, consideration can be made for tissue biopsy.

It may be of benefit to have patients with suggestive history referred to dedicated travel medicine clinics. Joint clinics involving infectious disease specialists and ophthalmologists would be beneficial for multi-disciplinary team management of relevant cases. In terms of preventive medicine, advisories should be made readily available for travellers to tick endemic areas to raise awareness of how to protect against or avoid tick bites.

#### Special Consideration: Atypical Presentations in the Immunocompromised and the Jarisch-Herxheimer Reaction (JHR)

It is known that immunocompromised hosts present with unusual or more severe manifestations of diseases. Specific to tick-borne diseases, Maraspin et al. reported that immunocompromised patients more commonly present with disseminated erythema migrans and extracutaneous manifestations of Lyme borreliosis [246]. Eiferman et al. reports a case of Borreliosis mimicking high-grade transformation of lymphoma in a patient with a background of lymphoma [247]. Other atypical, potentially life-threatening manifestations of borreliosis, anaplasmosis, and rickettsioses have been reported amongst transplant patients as well [248]. Thus, in such patients with suggestive exposure history, closer follow up and perhaps a lower threshold for workup is advised. It has been suggested that doses of antibiotics be augmented in the immunocompromised as well [249].

Similar to JHR, consideration should be made for immune reconstitution inflammatory syndrome (IRIS) which may result in unusual neurological presentations [250]. This syndrome has been commonly studied in patients with human immunodeficiency virus (HIV), and IRIS is known to occur in a significant number of susceptible patients within the first months of high active antiretroviral therapy (HAART) [251], with incidence varying depending on the underlying infection [252]. It is known to occur concomitantly with autoimmune and immune-mediated inflammatory disorders as well. Factors placing a patient at risk include a high pathogen load and low CD4+ T cell count on initiation of HAART. Rb-Silva et al. reported a case of central nervous system IRIS (CNS-IRIS) in a patient with HIV and toxoplasmosis infection, presenting with progressive hyposensitivity on routine follow up post-initiation of treatment [253]. In view of the lack of consensus regarding therapy for IRIS [254], a full dose anti-toxoplasma therapy along with HAART and opportunistic infection prophylaxis was maintained. No corticosteroids were administered. The patient subsequently improved. The report further reviewed treatment of five other cases of IRIS, four of whom received corticosteroid therapy with three having favourable outcomes [255,256] and one fatality [257]. Of the three with favourable outcomes, all three had recently been diagnosed with HIV/AIDS and had not been started on HAART prior to treatment for toxoplasmosis. The patient who died had a known history of AIDS with non-compliance to HAART prior to receiving therapy for toxoplasmosis. Both Cabral et al. and Rb-Silva et al. did not administer corticosteroids in their patients with favourable outcomes [253,258]. The pathogenesis of IRIS has yet to be elucidated fully, though it appears to depend upon a T cell-mediated response [259,260].

A corollary that may complicate treatment in even immunocompetent hosts is the Jarisch-Herxheimer reaction [261]. JHR is well-described in spirochaete disease and tick-borne diseases such as Lyme disease and relapsing fever. It may complicate clinicians' interpretation of treatment response in patients, resulting in dilemmas in choice of therapy [221,262,263]. The exact mechanism of JHR is unclear; however, it has been proposed that the breakdown of spirochaetes releases cytokines and toxins into the blood, triggering an acute inflammatory reaction. Although usually self-limiting, patients should be kept under close observation in view of potential for hypotension and end-organ damage.

There is a dearth of studies focusing on ocular manifestations related to tick-borne diseases in the immunocompromised population, certainly an area of consideration for future research given the susceptibility of the population and increasing prevalence of the disease.

*4.5. Diagnostic Dilemmas: Intraocular Malignancies*

Infectious agents are known to be mutagenic [264,265], although the pathway linking toxoplasmosis infection to carcinogenesis has yet to be established. A possible mechanism is a chronic stimulation of antigens of the host immune system. In murine models, DNA damage occurs with exposure to nitric oxide, γ-interferon, tumour necrosis factor-α (TNF-α), and reactive oxygen species (ROS) [266]. In Sauer et al.'s case report, *T. gondii* was present in ocular malignant cells but absent in ocular non-malignant cells, which suggests a relation between the infection and lymphogenesis [192]. Alternatively, lymphoma cells which proliferate more rapidly may be more susceptible to invasion as *T. gondii* has shown greatest affinity for host cell receptors during the S phase of the cell cycle [267].

It has also been suggested that the cholesterol of the host cell membrane influences entry of *T. gondii*. Coppens et al. found that depleting host cell plasma membrane cholesterol blocks entry of the parasite by reducing the release of rhoptry proteins by the parasite [268]. While lymphoma cells have lower cholesterol content than healthy lymphocytes [269], it is possible that cholesterol-rich rhoptries donate cholesterol to the host cell during the process of entry, though the hypothesis has yet to be studied.

Given that lymphoma is life-threatening, early recognition and treatment is paramount. The case report by Sauer et al. presents the possibility of toxoplasmosis being a potential

trigger for malignancy, though further studies are needed to define the pathway involved. An added quandary is the challenge in making the diagnosis of primary intraocular lymphoma.

## 5. Conclusions

The reviewed literature ranges from case reports and series to literature reviews of selected tick-borne illnesses. There are thus limits to the level of details provided regarding ophthalmic assessment, treatment rendered, and outcome of patients. In terms of investigations performed and choice of treatment, due to the variety of illnesses and time periods from which the articles were captured, there is heterogeneity in the management of patients.

The current literature demonstrates the wide range of manifestations of tick-borne diseases in the eye. With increased distribution of ticks, facilitated by climate change and travel, commensurate increase in awareness is key. This is especially so in regions non-endemic to ticks where tick-borne illnesses may be misdiagnosed at the outset. As discussed, the consequences of delayed diagnosis may be devastating. Although an extensive amount of literature exists in relation to ophthalmic manifestations of tick-related diseases, there is significant variation in clinical approaches and management of patients.

This article aims to provide a springboard for further research into the pathogenesis of various tick-borne illnesses and the development of a standardised clinical approach and algorithm to the evaluation and management of patients with possible tick-borne illnesses. Of further research interest is the possibility of tick-borne diseases as the catalyst for carcinogenesis of primary intraocular malignancies.

In conclusion, tick-borne diseases are a less recognised cause of ocular pathology that is worth remembering in the age of travel and climate change. Prompt recognition and treatment improves patient outcomes. Routine serologies for tick-borne illnesses are not recommended unless in areas endemic to the disease or in patients with red flags that increase suspicion of the diagnosis.

**Author Contributions:** Conceptualization, X.L.N. and C.H.L.L.; methodology, X.L.N. and C.H.L.L.; investigation, X.L.N., B.Y.Y.L. and C.X.C.C.; writing—original draft preparation, X.L.N., B.Y.Y.L. and C.X.C.C.; writing—review and editing, X.L.N., B.Y.Y.L. and C.X.C.C.; visualization, B.Y.Y.L. and C.X.C.C.; supervision, C.H.L.L., B.X.H.L. and D.K.A.L.; project administration, X.L.N. and C.H.L.L. All authors have read and agreed to the published version of the manuscript.

**Funding:** This research received no external funding.

**Institutional Review Board Statement:** Not applicable.

**Informed Consent Statement:** Not applicable.

**Data Availability Statement:** Not applicable.

**Conflicts of Interest:** The authors declare no conflict of interest.

# Appendix A

**Table A1.** Summary of the various ocular manifestations for the tick-borne diseases covered in this review and the treatment rendered.

| Tick-Borne Disease | Tick Species Involved | Ocular Manifestations | Duration to Onset | Treatment | Outcomes |
|---|---|---|---|---|---|
| Adnexal lesions | *Ixodes* spp. *I. nipponensis I. scapularis Dermacentor variabilis Rhipicephalus sanguineus Amblyomma americanum* | Eyelid ulceration, eyelid inflammation (acute or chronic), eyelid oedema, painful eyelid nodule ± mucopurulent discharge, conjunctival injection, conjunctival nodule, conjunctival hyperaemia, corneal precipitates, corneal thinning and vascularisation, palpebral ptosis, vasculitis. | Not stated. No history of tick bite [8]. Not stated [9,10]. Not stated. History of travel to a region endemic for tick bites [11]. 6 months after experiencing tick attachment in a wooded area (Cape Cod, Massachusetts, USA) [12]. Immediately after foreign body sensation in the right eye while camping (Adirondacks, New York, USA) [21]. 5 days after a hunting trip in a rural area of Alabama, USA [22]. 10 months after exposure to a cloud of unidentified insects in southern Spain [84]. | Topical tetracaine was applied, followed by irrigation with topical chlorhexidine. Tick was removed with toothed forceps. Topical tobramycin ointment was then applied. [8]. Tick was separated from the eyelid with 26G needle tip, then removed with toothed forceps. Prophylactic doxycycline 100 mg was given for 1 week against tick-borne diseases [9]. Tick was removed with toothless forceps [10]. Ticked was removed with blunt forceps, followed by topical chloramphenicol and dexamethasone sodium phosphate twice/day for 2 weeks. [11]. Complete excision of the suspicious nodule under local anaesthesia [12]. Topical proparacaine was applied, followed by removal of the tick with a 30G needle. Topical polymyxin-trimethoprim eye drops thrice/day and loteprednol eye drops twice/day were started for 3 days. PO doxycycline 100 mg given as prophylaxis against Lyme disease [21]. Topical proparacaine 0.5% was applied, followed by phenylephrine 2.5% topically near the organism. Jeweller's forceps were for elevation of conjunctiva. En bloc excision of the organism and surrounding conjunctiva was performed with Vannas scissors. Topical Bacitracin 4 times/day for 3 days was given [22]. Various antibiotic therapies were initially used, but had suboptimal results. Given irregular response to antibiotics and presence of vasculitis, azathioprine was started for immunosuppression, followed by topical cyclosporine to reduce surface inflammation [84]. | Full recovery [8,9,11,12,21,22,84]. Not stated [10]. |

**Table A1.** *Cont.*

| Tick-Borne Disease | Tick Species Involved | Ocular Manifestations | Duration to Onset | Treatment | Outcomes |
|---|---|---|---|---|---|
| Ophthalmic Lyme Disease | *Ixodes* spp. *I. scapularis* *I. pacificus* | **Lids** Erythema migrans on lids and periocular adnexae Palpebral oedema Blepharospasm. **Conjunctiva** Conjunctivitis (follicular) Symblepharon Subconjunctival haemorrhages. **Episcleritis** **Scleritis** **Cornea** Keratitis (exposure, interstitial, peripheral ulcerative, stromal) (associated with peripheral stromal oedema and mild corneal neovascularisation which is infrequent) Cogan's syndrome. **Pupils** (Reversible) Horner's syndrome Argyll Robertson Afferent pupillary defect Tonic pupils. **Cranial Nerve Palsies** CN3, 4, 6, 7 Paralytic strabismus. **Uvea** Iritis Cyclitis Iridocyclitis Choroiditis: Chorioretinitis, multifocal choroiditis, Birdshot chorioretinopathy, Acute posterior multifocal placoid pigment epitheliopathy Uveitis: Anterior, Intermediate (most common), Posterior, Panuveitis (granulomatous, associated with anterior synechiae) Posterior synechiae Inflammatory choroidal neovascular membrane. **Vitreous** Vitritis (anterior "spiderweb" vitritis without retinal involvement) Atypical pars planitis syndrome Pars planitis Vitreous clouding. | 3 cases were described by Smith et. al. Case 1: The patient presented with blurred vision after 3 years of multiple facial palsies that were responsive to steroids. The patient had a tick bite while hunting in Everglades several years before and camped at Cape Cod 11 and 14 years prior to presentation. Case 2 and 3: not stated [81]. 4 weeks after the patient had *Erythema chronicum migrans* [83]. Sauer et. al. described 2 cases. Case 1: Current complication of horizontal diplopia developed after 3 years of recurrent episodes of right orbital swelling and pain. Case 2: Recent history of tick bite followed by *Erythema migrans* (no specific duration stated) [88]. 6 months after tick bite [89]. The patient was hunting and skinning deer in Southeast Pennsylvania the month before presentation and noticed a large tick on his neck 4 days prior to presentation [103]. Presented with a scotoma above fixation in the left eye after 2 months of floaters in both eyes and daily throbbing headaches [112]. Not stated [116]. The patient presented after a 1 month history of a dull temporal headache which was treated by a chiropractor [124]. | Successful treatment with topical Prednisolone acetate 1% every 4 h [46]. Topical corticosteroid therapy with prednisolone sodium phosphate (1%) four times/day, later started on PO doxycycline 100 mg twice/day for 21 days [47]. Successful treatment with Prednisolone acetate 1% eye drops four times daily [48]. Prednisone 60 mg was given with ocular complications developing after: rise in intraocular pressure, proptosis, conjunctival purulent discharge and rapid onset of a dense cataract [54]. PO doxycycline 100 mg twice/day for 7 days [67]. Systemic IV ceftriaxone for neuroretinitis [73]. Case 1, during a recurrence of retinal vasculitis: PO tetracycline 250 mg four times/day, 20 mg subtenon Kenalog (aqueous triamcinolone). Case 2: IV aqueous penicillin 20 million units daily. Case 3: PO Isoniazid for positive PPD skin test, subtenon Depomedrol injections, panretinal photocoagulation for increasing vitreous haemorrhage and neovascularisation, posterior vitrectomy with membrane peeling due to increasing traction and blood-obscuring laser treatment, IV Rocephin 2 g/day for 2 weeks at outpatient [81]. Treatment with corticosteroids failed to provide improvement, IV methicillin 12 g/day and IV gentamicin 80 mg every 8 h were then started after positive Lyme disease serology but vision worsened, followed by lensectomy and vitrectomy with repeated drainage of purulent vitreous debris and the administration of intravitreal gentamicin 0.2 mg and chloramphenicol 0.2 mg [83]. Case 1: PO doxycycline 200 mg/day. Case 2: Doxycycline 200 mg/day for 4 weeks [88]. IV ceftriaxone 1500 mg/day for 3 weeks [89]. After diagnosis of Lyme disease, was originally treated with PO tetracycline 500 mg 4 times/day, but then noticed unequal pupils and a drooping left lid. Following that, he was treated with 1% hydroxyamphetamine (Paredrine) and IV Ceftriaxone 1 g every 12 h for 10 days [103]. IV penicillin G 12 million units daily for 10 days [112]. | Uneventful full recovery [46,47,67, 88,89,103,116]. Treatment resulted in recovery, but there was 1 recurrence which also resolved with similar therapy. No further recurrences occurred [48]. Developed dense cyclitic membrane in the eye and lost all functional visual activity, with the eye becoming phthisical. The patient was later given a cosmetic contact lens shell covering the phthisical left eye [54]. Case 1: Patient previously had multiple ocular manifestations which were resolved with a combination of antibiotics. For this particular recurrence of retinal vasculitis: the patient recovered well with the described treatment. Case 2: No follow-up was available for this patient since his discharge Case 3: Stabilised with no recurrence [81]. Eye became phthisical and all vision was lost [83]. |

**Table A1.** *Cont.*

| Tick-Borne Disease | Tick Species Involved | Ocular Manifestations | Duration to Onset | Treatment | Outcomes |
|---|---|---|---|---|---|
| Ophthalmic Lyme Disease | *Ixodes* spp. *I. scapularis* *I. pacificus* | **Retina** Retinitis Retinal vasculitis Atypical Eales disease syndromes Exudative retinal detachments Branch retinal artery occlusion with cotton wool spots Horseshoe-shaped retinal tear with inflammatory nodules on the flap Macular oedema Retinal venular occlusions: branch retinal vein occlusion Chorioretinal inflammatory foci Secondary retinitis pigmentosa Pigment epitheliitis. **Optic Nerve** Optic neuritis (retrobulbar neuritis) Optic perineuritis Neuroretinitis (Leber's stellate) complications: full thickness macular hole and peripapillary retinal pigment epithelium detachments) Chiasmal optic neuritis Papilloedema (possibly associated with meningitis) Ischaemic optic neuropathy (anterior) Big blind spot syndrome Secondary optic atrophy. **Orbit** Orbit periostitis Periorbital oedema Orbital myositis (medial rectus, inferior rectus, lateral rectus), associated with lacrimal gland enlargement and optic nerve sheath contrast enhancement on MRI. **Others** Endophthalmitis Panophthalmitis Cortical blindness Intraocular inflammatory syndromes Pseudotumour cerebri Photophobia Acute visual loss Opsoclonus (Opsoclonus-myoclonus syndrome) Nystagmus (associated with partial CN VI palsy). **Temporal Arteritis** | | Ceftriaxone 2 g/day [116]. Originally treated with prednisone 80 mg/day but the patient deteriorated. The patient was then given IV Decadron 8 mg every 6 h for 24 h which stabilised the vision in his nonamblyopic left eye. This was followed by his discharge and tapering of the corticosteroids. After a *Borrelia*-compatible spirochaete (but not *B. burgdorferi*) was identified in peripheral blood cultures, IV ceftriaxone sodium 2 g daily over 7 days was administered and a repeat peripheral blood culture yielded no evidence of spirochaetes. A 2nd course of steroids failed to improve the patient's vision (counting fingers at 5 feet) [124]. | Systemic symptoms and floaters improved with the fundi returning to normal, but the scotoma that the patient first presented with persisted and the patient's VA remained unchanged [112]. Vision remained at counting fingers at 5 feet, loss of vision in the patient's nonamblyopic left eye necessitated the patient to suddenly retire from dentistry [124]. |

**Table A1.** *Cont.*

| Tick-Borne Disease | Tick Species Involved | Ocular Manifestations | Duration to Onset | Treatment | Outcomes |
|---|---|---|---|---|---|
| Tularaemia | *Dermacentor* spp. *D. variabilis* *D. andersoni* *Amblyomma americanum* | **Oculoglandular tularaemia** Conjunctival chemosis, episcleritis, conjunctivitis, ptosis, purulent secretions, periorbital oedema, conjunctival injection and hyperaemia, uveitis, conjunctival papule, conjunctival ulcer. **Atypical for oculoglandular tularaemia** Corneal oedema, raised intraocular pressure, dacryocystitis. *Parinaud's oculoglandular syndrome* Associated with preauricular and submandibular lymphadenopathy. Conjunctivitis, periorbital ecchymosis, conjunctival nodules ± ulcers, eyelid oedema, conjunctival follicles with mucous discharge, corneal ulceration, hypopyon. | **Tularaemia** Case 1: 3 days after cat sneezed and secretions were projected into patient's eye. Case 2: few days after contact with an ill cat. Case 3: few days after contact with an ill puppy [128]. 3 to 5 days after exposure to contaminated substances [129]. 3 weeks after exposure to an infected rabbit [130]. Few days, no history of tick bite. [131]. Not stated [132,138–140]. 2 days after laparotomy for lung biopsy [141]. **Parinaud's oculoglandular syndrome** 3 weeks after contact with dead wild rabbit [125]. 5 days after contact with contaminated sewage water and tick bites [134]. Not stated [135]. 4 to 5 days after tick bite [137]. | **Tularaemia** Case 1: local cold boric compresses, triple-typhoid bacterial IV vaccine. Case 2: hourly instillations of 20% silver iodide followed by cold boric-acid compresses. Case 3: metaphen instillations, cold boric compresses [128]. Aminoglycoside or fluoroquinolone antibiotics for at least 10 days, or doxycycline for at least 15 days. Local therapy included ciprofloxacin and tobramycin eye drops/ointment [129]. Doxycycline 200 mg once/day, for 3 weeks [130]. Initially started on streptomycin and tetracycline, later switched to tetracycline-only targeted antibiotic therapy [131]. IM streptomycin 65 mg/kg/day twice/day for 7 days [132]. Raised intraocular pressure was lowered with IV and PO acetatcolamide, timolol and pilocarpine eyedrops. Laser iridotomy was done due to a narrow anterior chamber. Ciprofloxacin 500 mg twice/day for 10 days [138]. PO amoxicillin-clavulanic acid 1000 mg twice/day for 14 days and gentamicin eye drops. Surgical drainage for dacryocystitis [139]. PO amoxicillin-clavulanic acid 1 g twice/day and topical ciprofloxacin eye drops 4 times/day [140]. Gentamicin sulfate and tetracycline hydrochloride [141]. **Parinaud's oculoglandular syndrome** IM streptomycin and IV nafcillin [125]. 10-day course of gentamicin [134]. PO doxycycline 100 mg every 12 h for 10 days [135]. Doxycycline hyclate 100 mg twice/day for 3 weeks and 14 mg/mL topical fortified gentamicin sulfate eye drops hourly to affected eye [137]. | Not stated [128,129, 131,132,134,135,140]. Visual loss with central scotoma [130]. Required corneal transplantation due to persistent medial opacities [137]. Intraocular pressure of the right eye decreased from 68 mmHg at first presentation to 26 mmHg at the half-year mark follow-up. Vision was normal [138]. Full recovery [139]. |

**Table A1.** *Cont.*

| Tick-Borne Disease | Tick Species Involved | Ocular Manifestations | Duration to Onset | Treatment | Outcomes |
|---|---|---|---|---|---|
| Babesiosis | *Ixodes scapularis* | Conjunctival injection, retinal haemorrhages, retinal nerve fibre layer infarct, conjunctivitis, eyelid oedema. | 2 months after visiting Shelter Island in New York [142]. 3 months after a blood transfusion [143]. Few months after a blood transfusion [144]. | Not stated [142]. Doxycycline 200 mg/day and trimethoprim-sulfamethoxazole 160/800/per day [143]. Extensive doses of penicillin, hydrocortisone and peritoneal dialysis [144]. | Full recovery [142,143]. Death [144]. |
| Tick-borne Relapsing Fever | *Ornithodoros* spp. *O. hermsi* *O. parkeri* *O. turicata* | Anterior uveitis, intermediate uveitis, optic neuritis, choroiditis, vitritis, endophthalmitis, floaters, acute iridocyclitis, chronic cyclitis, posterior synechiae, marginal keratitis followed by multiple corneal erosions. | Case 1: 2 months after generalised aches and pains with recurrent episodes of fever, 1 week after an episode of relapsing fever. Case 2: 2 months after pain in head and eyes associated with relapsing fever. Case 3: Not stated. Case 4: 4 months after relapsing fever, with the most recent month having no relapse [147]. Systemic illness occurred 10 days after the patient had been camping in a forest cabin in Eastern Oregon [149]. | For posterior synechiae: mydriatic agents (atropine), 2% cocaine eye drops and heat. For marginal keratitis: 2% silver nitrate [147]. IV Ceftriaxone 1 g twice/day for 3 days, followed by PO cephuroxime 250 mg twice/day for 4 weeks was given for presumed sinusitis noted on computer tomography. Symptoms resolved but the patient developed floaters and blurred vision OD afterwards. Topical prednisolone acetate was given but inflammation persisted, then doxycycline 100 mg/day for 4 weeks was added on to the corticosteroid eye drops for presumed residual infection [149]. | Uneventful full recovery [147,149]. |
| Powassan Encephalitis | *Ixodes* spp. *I. cookei* *I. marxi* *I. spinipalpis* *I. scapularis* *Dermacentor andersoni* | Retinal vein tortuosity, optic disc oedema, ophthalmoplegia, multifocal choroiditis. | Few days after initial symptoms of nausea and vomiting, diarrhoea, dizziness, diplopia and incoordination [196]. | Combination of ceftriaxone, ampicillin and acyclovir [196]. | Residual ophthalmoplegia [196]. |
| Ehrlichiosis | *Amblyomma americanum* *Dermacentor variabilis* | CN IV palsy, optic neuritis, disc oedema, orbital myositis, posterior uveitis, cystoid macular oedema, retinal vasculitis, epiretinal membrane. | 12 days after constitutional symptoms of fever, chills, myalgia and malaise [153]. 18 days after initial systemic symptoms [154]. 6 weeks after constitutional symptoms of high fever, myalgia and arthralgia [155]. 1 month after subjective deterioration in vision [156]. | Antibiotics and corticosteroids—oral doxycycline, sub-Tenon triamcinolone injection. PO doxycycline [153]. PO doxycycline 100 mg twice/day [154]. PO doxycycline 100 mg twice/day [155]. Sub-Tenon triamcinolone for both eyes and a prophylactic course of PO doxycycline 100 mg twice/day for 2 weeks [156]. | Full recovery [153–155]. Clinical improvement of macular oedema [156]. |

**Table A1.** *Cont.*

| Tick-Borne Disease | Tick Species Involved | Ocular Manifestations | Duration to Onset | Treatment | Outcomes |
|---|---|---|---|---|---|
| Rocky Mountain Spotted Fever | *Dermacentor* spp. *D. variabilis* *D. andersoni* *Rhipicephalus sanguineus* | Conjunctival vasculitis, retinal vasculitis, choroidal vasculitis, conjunctival injection, periorbital oedema, uveitis, keratic precipitates, anterior chamber and vitreous cells. **Retinovascular** Cotton wool spots, cotton wool exudates, macular oedema, retinal oedema, vascular sheathing, venous tortuosity, retinal artery or vein occlusion, intraretinal haemorrhage and exudates, retinitis, retinal artery sheathing, flame-shaped haemorrhages, macular star figures. **Neuro-ophthalmic** Disc oedema, optic neuritis, optic neuropathy, neuroretinitis, optic nerve oedema, papilloedema. | Presley reported 6 cases. Case 1: Not stated. Case 2: 10 days after a tick was removed from the patient's scalp. Case 3: 1 week after the patient removed 2 ticks from his body, and another tick was removed on the day of admission. Case 4: 2 weeks after a tick was removed from the patient's scalp. Case 5: 6 days after the patient's father had removed an unattached tick from the patient. Case 6: A history of possible tick bite 2 weeks prior to admission [164]. 10 days after tick bite [165]. 5 days after travel to Mexico, without exposure to stray animals, insects or tick bites [166]. 2 weeks after tick bite [167]. Several months after sustaining tick bites [168]. Presented after a 1-week history of papular skin lesions, associated with 2 weeks of systemic symptoms. She also lived alone with her dog in a Philadelphia tenement. [169]. | Not stated [164]. IV chloramphenicol, PO tetracycline [165]. 14 days of doxycycline [166]. PO doxycycline 100 mg every 12 h for 60 days [167]. PO doxycycline 100 mg twice/day for 14 days [168]. Supportive therapy (IV fluids), IV tetracycline [169]. | Not stated [164]. Uneventful full recovery [165,166,169]. Slow and incomplete recovery of vision (improvement in visual acuity and visual fields) with resolution of optic disc swelling and development of optic disc pallor [167]. Improvement of optic nerve oedema with VA remaining at 20/40 OD and improvement of VA to 20/100 OS. Macular star figures persisted [168]. |
| Mediterranean Spotted Fever | *Rhipicephalus sanguineus* | Parinaud's oculoglandular syndrome, corneal manifestations, posterior segment manifestations, dacryoadenitis, multifocal retinitis. **Parinaud's oculoglandular syndrome** Conjunctivitis—swollen eyelids, conjunctival hyperemia, and chemosis with mucopurulent discharge. **Corneal Manifestations** Keratitis, corneal ulcers (ameboid-type), ciliary injection, corneal oedema, corneal inflammatory infiltrates, mild infiltration of the anterior stroma. | Average duration of fever before ophthalmic examination was 7 days (range, 3–15). For the 9 (out of 30) patients who had ocular complaints: interval from the onset of fever to ocular symptoms ranged from 2 to 5 days [174]. 4 weeks after the patient had an accidental projection of a jet of contaminated water into his left eye. Could not recall any history of tick bite [175]. | PO doxycycline for 2 weeks [174]. PO doxycycline 100 mg twice/day for 2 weeks [175]. PO doxycycline 100 mg/12 h, while keratitis was treated with atropine 1% plus tetracycline ointment every twelve hours and occlusion [178]. | One eye had retinal neovascularization at the 6-month follow-up, but a further follow-up of 6 months reported no other abnormalities. All posterior segment findings at the acute stage resolved in 3 to 10 weeks; final VA was 20/20 in 42 of 45 affected eyes; a decreased final VA was related to RPE changes due to cystoid macular oedema (1 eye) and age-related cataract (2 eyes) [174]. |

**Table A1.** *Cont.*

| Tick-Borne Disease | Tick Species Involved | Ocular Manifestations | Duration to Onset | Treatment | Outcomes |
|---|---|---|---|---|---|
| Mediterranean Spotted Fever | *Rhipicephalus sanguineus* | **Posterior Segment Manifestations** Acute anterior ischemic optic neuropathy, RAPD, retinal vasculitis, haemorrhages (optic disc, intraretinal, white-centred retinal, subretinal), branch retinal artery and vein occlusions, retinal detachment (serous), optic disc oedema, cotton-wool spots and retinitis with mild vitreous inflammatory reaction, optic disc staining, juxtavascular white retinal lesions, focal vascular sheathing, multiple arterial plaques, macular star, cystoid macular oedema, retinal vascular leakage, delayed filling in a branch retinal vein, multiple hypofluorescent choroidal dots. Ocular complaints: decreased vision, paracentral scotoma, floaters and ocular redness due to conjunctivitis or anterior uveitis. | 14 days after constitutional symptoms. Tick bite present on left leg [178]. 4 days after the onset of a fever. No history of tick bite [179]. | Doxycycline 200 mg/day for a week [179]. | Uneventful recovery [175,178]. Clinical improvement: BCVA counting fingers OD and 20/20 OS increased to 20/400 OD and 20/20 OS. Optic disc oedema OD was replaced by pallor, otherwise the patient recovered [179]. |
| Colorado Tick Fever | *Dermacentor andersoni* | Retro-orbital pain, photophobia and conjunctival injection. | NIL | Supportive treatment [199]. | NIL |

**Table A1.** *Cont.*

| Tick-Borne Disease | Tick Species Involved | Ocular Manifestations | Duration to Onset | Treatment | Outcomes |
|---|---|---|---|---|---|
| Toxoplasmosis | Studies have suggested the **possibility** of transmission of *Toxoplasma gondii* through the following ticks: *Dermacentor variabilis*, *Dermacentor andersoni*, *Amblyomma americanum*, *Dermacentor reticulatus*, *Ixodes ricinus*, *Amblyomma cajennense* complex, mainly *Amblyomma sculptum*, *Ornithodorus moubata* and *Haemaphysalis longicornis* | **Posterior uveitis.** Particularly focal retinochoroiditis. Less commonly: serous macular detachment, retinal vasculitis, retinal detachment, neuroretinitis, papillitis, disc haemorrhages with venous engorgement, optic atrophy secondary to optic nerve involvement, macular star, cystoid macular oedema, chronic iridocyclitis, cataract formation, secondary glaucoma, band keratopathy. **Association with primary intraocular lymphoma.** Focal retinitis with surrounding retinal and macular pigmentary changes, vitritis, cells and flare of anterior chamber and vitreous cells. | 2 month history of foggy vision without systemic symptoms [192]. | Recommended: A combination of antiprotozoal agents, systemic steroids if no contraindication, topical steroids and cycloplegics if anterior segment inflammation is present. Initial treatment was for presumed ocular toxoplasmosis with trimethoprim 160 mg and sulfamethoxazole 800 mg twice/day which resolved the focal retinitis. Patient later had recurrences of increased floaters and blurred vision, and was restarted on antibiotics. Systemic chemotherapy was instituted upon diagnosis of lymphoma [192]. | Patient did not respond to chemotherapy and died shortly thereafter [192]. |
| Tick-Borne Encephalitis | *Ixodes* spp. *I. ricinus* *I. persulcatus* | Non-granulomatous anterior uveitis, flame-shaped and dot retinal haemorrhages, vitritis, vitreous haze and cells. | 6 weeks after a tick bite, where he experienced a flu-like illness 10 days later but no erythema migrans [197]. | Topical prednisolone acetate 1% 6 times/day for anterior uveitis. Empirical IV acyclovir 10 mg/kg thrice/day was initiated for possible herpes meningoencephalitis with retinal involvement [197]. | Full recovery [197]. |

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
