# Peer review of "Revenge of the Tick: Tick-Borne Diseases and the Eye in the Age of Climate Change and Globalisation"

_zoonoticdis, doi:10.3390/zoonoticdis2040017_

Round 1

Reviewer 1 Report

Although it is a little difficult to follow due to the multiplicity of the headings and subheadings, it has been a very comprehensive and extensive review.

Author Response

Thank you. We have included 2 new tables for greater readability - Table 1 that tabulates the diseases, agents and transmission, and Table 2 that encapsulates the reported manifestations and site of ocular involvement.

Reviewer 2 Report

Congratulation on all the work that you have done. The manuscript is well established. I have only some suggestions for being more easy reading.

L 54-91: you can have a table with the diseases, agents, and transmission, instead of this part.

The results should be written in a table, except in the manuscript. It is quite chaotic. They should be categorized either based on the agent or based on the ocular manifestation. In my opinion, it is better to be categorized based on the ocular manifestation, as the symptom can help with the differential diagnosis. Anyway, there should be tables, in order to help the reading, as the information is too much. 

Author Response

Congratulation on all the work that you have done. The manuscript is well established. I have only some suggestions for being more easy reading.

Response: We thank the reviewer for their kind comments and suggestions for improvements, which have been incorporated into the revised manuscript.

L 54-91: you can have a table with the diseases, agents, and transmission, instead of this part.

Response: We have included Table 1 at the end of the Introduction section to summarise the text in L54-91 for greater ease of reading and comprehension.

The results should be written in a table, except in the manuscript. It is quite chaotic. They should be categorized either based on the agent or based on the ocular manifestation. In my opinion, it is better to be categorized based on the ocular manifestation, as the symptom can help with the differential diagnosis. Anyway, there should be tables, in order to help the reading, as the information is too much.

Response: We have included Table 2 at the beginning of the Results section. This summarises the reported ocular manifestations and site of ocular involvement.

Reviewer 3 Report

It's a good review on diseases associated with tick bites, however the work is 90% Lyme. I am not able to conclude that climate change is responsible for the expansion of tick species and ophthalmic diseases. The diversity of tick species is indeed related to environmental factors and hosts, however, I was not able to visualize in the manuscript how much this diversity provides an increase in ophtalmic diseases and tick bites, except for Lyme Borreliosis. So I think the conclusions are unclear.

Other corrections were made directly to the text, especially scientific names of ticks.

Author Response

It's a good review on diseases associated with tick bites however the work is 90% Lyme. I am not able to conclude that climate change is responsible for the expansion of tick species and ophthalmic diseases. The diversity of tick species is indeed related to environmental factors and hosts, however, I was not able to visualize in the manuscript how much this diversity provides an increase in ophtalmic diseases and tick bites, except for Lyme Borreliosis. So I think the conclusions are unclear.

Response: We thank the reviewer for their kind comments and suggestions for improvements, which have been incorporated into the revised manuscript .

Under Section 4.3: Climate Change and Air Travel, we have further elaborated on the possible mechanisms and provided examples on how climate change relates to the expansion of tick species to previously non-endemic areas. Particularly, we have used Ixodes ricinusand Dermacentor reticulatus (which are ticks associated with Tick-Borne Encephalitis, Lyme Borreliosis, Ehrlichiosis and Toxoplasmosis) as case examples to illustrate how various aspects of climate change have affected the distribution of tick-borne diseases.

Other corrections were made directly to the text, especially scientific names of ticks.

Response: We have also edited the scientific names of ticks in the manuscript as kindly recommended by the reviewer.